# Downregulation of chemokine receptor 9 facilitates CD4+CD8αα+ intraepithelial lymphocyte development

Keiko Ono[1,11], Tomohisa Sujino [2]✉, Kentaro Miyamoto[1,3,11], Yosuke Harada[1,11], Satoshi Kojo [4,10], Yusuke Yoshimatsu[1], Shun Tanemoto[1], Yuzo Koda [1,5], Jiawen Zheng[4], Kazutoshi Sayama[6], Tsuyoshi Koide[7], Toshiaki Teratani[1], Yohei Mikami [1], Kaoru Takabayashi[2], Nobuhiro Nakamoto [1], Naoki Hosoe[2], Mariya London[8], Haruhiko Ogata[2], Daniel Mucida [8,9], Ichiro Taniuchi [4] & Takanori Kanai[1]✉

Intestinal intraepithelial lymphocytes (IELs) reside in the gut epithelial layer, where they help in maintaining intestinal homeostasis. Peripheral CD4+ T cells can develop into CD4+CD8αα+ IELs upon arrival at the gut epithelium via the lamina propria (LP). Although this specific differentiation of T cells is well established, the mechanisms preventing it from occurring in the LP remain unclear. Here, we show that chemokine receptor 9 (CCR9) expression is low in epithelial CD4+CD8αα+ IELs, but CCR9 deficiency results in CD4+CD8αα+ over-differentiation in both the epithelium and the LP. Single-cell RNA sequencing shows an enriched precursor cell cluster for CD4+CD8αα+ IELs in *Ccr9*[−/−] mice. CD4+ T cells isolated from the epithelium of *Ccr9*[−/−] mice also display increased expression of Cbfβ2, and the genomic occupancy modification of Cbfβ2 expression reveals its important function in CD4+CD8αα+ differentiation. These results implicate a link between CCR9 downregulation and *Cbfb2* splicing upregulation to enhance CD4+CD8αα+ IEL differentiation.

The intestine is exposed to both food-derived and microbiota-derived antigens, which are separated from the core of the body by a single layer of epithelial cells. Within the epithelium, there are various subsets of intraepithelial lymphocytes (IELs) that are crucial for maintaining the mucosal barrier[1]. IELs comprise a heterogeneous cell population, and various IEL subsets are distributed in the intraepithelial (IE) compartment[1]. CD4+CD8αα+ IELs, a subset of IELs that express CD4 and CD8αα, exhibit regulatory properties against intestinal inflammation. CD4+CD8αα+ IELs are differentiated from CD4+CD8αα− IELs or ex-Foxp3+ T cells (ex-Treg cells) upon migration to the IE[2–5]. Although most regulatory T cells (Treg cells) reside in the lamina propria (LP), CD4+CD8αα+ IELs are located in the IE compartment.

Retinoic acid (RA) promotes the migration of CD4+ T cells that have encountered antigens to the small intestine by upregulating the expression of chemokine receptor 9 (CCR9) in CD4+ T cells[6,7]. This

---

[1]Division of Gastroenterology and Hepatology, Department of Internal Medicine, Keio University School of Medicine, Tokyo, Japan. [2]Center for Diagnostic and Therapeutic Endoscopy, Keio University School of Medicine, Tokyo, Japan. [3]Research Laboratory, Miyarisan Pharmaceutical Co., Tokyo, Japan. [4]Laboratory for Transcriptional Regulation, RIKEN Center for Integrative Medical Sciences, Yokohama, Japan. [5]Mitsubishi Tanabe Pharma Corporation, Kanagawa, Japan. [6]Applied Life Science Course, College of Agriculture, Shizuoka University, Shizuoka, Japan. [7]Mouse Genomics Resource Laboratory, National Institute of Genetics, Shizuoka, Japan. [8]Laboratory of Mucosal Immunology, The Rockefeller University, New York, NY 10065, USA. [9]Howard Hughes Medical Institute, The Rockefeller University, New York, NY 10065, USA. [10]Present address: Division of Immunology and Stem Cell Biology, Institute of Medical, Pharmaceutical and Health Science, Kanazawa University, Kanazawa, Japan. [11]These authors contributed equally: Keiko Ono, Kentaro Miyamoto, Yosuke Harada. ✉e-mail: tsujino1224@keio.jp; takagast@keio.jp

migration to the small intestine is facilitated by the high expression of C-C motif chemokine ligand 25 (CCL25), the ligand for CCR9[7,8]. CCR9+CD4+ T cells migrate into the LP of the small intestine. Upon arrival, some cells move back and forth between the LP and the IE compartment, ultimately residing in the IE compartment. After this migration, activated T cells gradually express αEβ7 (CD103) and interact with E-cadherin expressed by enterocytes[1]. Factors enriched in the intestinal milieu, such as interferon-γ (IFN-γ), transforming growth factor-β (TGF-β), RA, and microbiota species, including *Lactobacillus reuteri*, activate the epithelium-specific differentiation program in CD4+ T cells[2–4,9–11]. One important step in CD4+CD8αα+ IEL differentiation is the loss of CD4 lineage–defining transcription factor ThPOK and the upregulation of CD8 lineage–defining transcription factor Runx3[3]. Runx3 functions as a heterodimer with one of two splice variants, Cbfβ1 or Cbfβ2, generated from *Cbfb*, each with a distinct C-terminal amino acid sequence[12,13]. Cbfβ2 provides thymic homing capacity to pre-thymic progenitors by inducing the expression of CCR9[14]. In turn, the expression of both CCR9 and α4β7 integrin in T cells confers LP residency[6,15,16]. However, whether Cbfβ1 or Cbfβ2 plays distinct functions in regulating CD4+CD8αα+ IEL differentiation has not been elucidated. Furthermore, it remains unclear whether and how CCR9 expression changes during T-cell migration and differentiation from the LP into the IE compartment. Overall, the mechanism by which peripheral CD4+ T cells acquire an IEL profile during migration via the LP remains poorly understood.

In the present study, we discover that CCR9 expression is downregulated in CD4+CD8αα+ IELs compared to that in CD4+CD8αα− IELs and Treg cells in the IE compartment. Conversely, CD4+CD8αα+ IELs are more abundant in *Ccr9−/−* mice than in *Ccr9+/+* mice. However, we also found that the CCR9 ligand, CCL25, did not interfere with CD4+CD8αα+ IEL differentiation. Single-cell RNA sequencing (scRNA-seq) of IELs from *Ccr9−/−* and *Ccr9+/+* mice shows that CD4+ T cells in the epithelia of *Ccr9−/−* mice exhibits greater expression of *Runx3, Tbx21*, and *Cbfb*, which are essential for the development of CD4+CD8αα+ IELs, than those of *Ccr9+/+* mice. *Cbfb* has two splicing forms, *Cbfb1* and *Cbfb2*, and we observed that *Cbfb2*, but not *Cbfb1*, is more highly expressed in CD4+ IELs from *Ccr9−/−* mice than in those from *Ccr9+/+* mice. Additionally, transgenic expression of *Cbfb2* increases the differentiation of CD4+CD8αα+ IELs in vivo. In summary, our findings suggest that CCR9 downregulation in CD4+ T cells facilitates their differentiation towards epithelium-adapted CD4+CD8αα+ IELs via upregulation of the Runx3-partner *Cbfb2*.

## Results

### CCR9 is downregulated during CD4+ IEL differentiation

Within IEL subsets, CD4+ T cells can be divided into three populations (Supplementary Fig. 1a): CD4+CD8αα+, CD4+CD8αα−, and Treg cells. Because CCR9 and α4β7 represent the main gut-homing receptors, we analyzed their expression in the spleen, mesenteric lymph nodes (MLN), and small intestine layers to identify possible expression patterns during gut homing. Interestingly, the expression of both receptors was lowest in CD4+CD8αα+ IELs (Fig. 1a, b and Supplementary Fig. 1b–d). To assess gene expression in differentiating intestinal CD4+ T cell populations, we used *Thpok*GFP:*Runx3*tdTomato reporter mice[3] and sorted intraepithelial CD4+ T cells into ThPOKhiRunx3low (less differentiated), ThPOKhiRunx3hi (transitioning), and ThPOKlowRunx3hi (more differentiated). *Ccr9* mRNA and protein expression was higher in ThPOKhiRunx3low IELs than in ThPOKlowRunx3hi IELs; *Tbx21* and *Runx3* expression was higher in ThPOKlowRunx3hi IELs than in ThPOKhiRunx3low IELs (Fig. 1c–e). Furthermore, ThPOKhiRunx3low cells in the lamina propria lymphocytes (LPLs) exhibited higher CCR9 expression than ThPOKlowRunx3hi cells (Supplementary Fig. 1e, f). These data indicated that downregulation of CCR9 accompanied the CD4+CD8αα+ T cell differentiation process.

### The development of CD4+CD8αα+ T cells does not require CCL25 signaling

To investigate whether the interaction between CCR9 and CCL25 affects CD4+CD8αα+ IEL development, we evaluated the expression levels of *Ccl25* in the small intestine. We found that *Ccl25* mRNA expression was higher in the ileum than in the jejunum (Supplementary Fig. 2a), but the proportion of CD4+CD8αα+ IELs was comparable in both regions (Supplementary Fig. 2b). Additionally, the expression of CCR9 among CD4+CD8αα+, CD4+CD8αα−, and Treg cells of IELs was cell-type-dependent and not influenced by the location in the small intestine (Fig. 1a, b). To clarify whether CCL25 is involved in the development of CD4+CD8αα+ IELs, we analyzed *Ccl25+/+* and *Ccl25−/−* mice. The proportion of CD4+CD8αα+ IELs was comparable between *Ccl25+/+* and *Ccl25−/−* mice (Supplementary Fig. 2c). Taken together, these results suggested that CCL25 was not involved in the induction of CD4+CD8αα+ IELs.

To investigate whether CCL25 acts independently of CCR9 expression as a modulator of CD4+CD8αα+ IELs, we then analyzed CCR9 expression in the IELs of *Ccl25+/+* and *Ccl25−/−* mice. The MFI of CCR9 in Treg cells, CD4+CD8αα− IELs, and CD4+CD8αα+IELs of *Ccl25−/−* mice was higher than that in each population of *Ccl25+/+* mice (Supplementary Fig. 2d, e). Notably, similar to our observations in WT mice, the MFI of CCR9 in Treg cells from *Ccl25−/−* mice was higher than that in CD4+CD8αα− IELs and CD4+CD8αα+ IELs from *Ccl25−/−* mice. Similarly, the MFI of CCR9 in CD4+CD8αα− IELs from *Ccl25−/−* mice was higher than that in CD4+CD8αα+ IELs from *Ccl25−/−* mice. Hence, while the expression level of CCR9 was high in *Ccl25−/−* mice due to the lack of CCL25; however, the patterns of CCR9 expression observed in WT mice were recapitulated in *Ccl25−/−* mice. These data suggest downregulation of CCR9 accompanied the CD4+CD8αα+ T cell differentiation process in CCL25-independent manner.

### Loss of CCR9 enhances CD4+CD8αα+ IEL differentiation

To investigate the impact of CCR9 on CD4+CD8αα+ IEL differentiation, we analyzed *Ccr9−/−* and CCR9-transgenic mice. Although the frequency of Treg cells and T helper (Th) 1 cells in the LP was similar between *Ccr9+/+* and *Ccr9−/−* mice at 10 weeks of age, the frequency of Th17 cells in the LP was higher in *Ccr9−/−* mice than in *Ccr9+/+* mice (Supplementary Fig. 3a). The total number of IELs and CD4+ IELs was comparable between *Ccr9+/+* and *Ccr9−/−* mice (Supplementary Fig. 3b). Notably, the frequency and abundance of CD4+CD8αα+ IELs within the CD4+ T cell subset was higher in *Ccr9−/−* mice than in age-matched *Ccr9+/+* mice (Fig. 2a–c). Furthermore, the frequency and abundance of CD4+CD8αα+ T cells in the LP was also higher in *Ccr9−/−* mice than in *Ccr9+/+* mice (Supplementary Fig. 3c).

Because the development of CD4+CD8αα+ IELs and Th17 cells is microbiota-dependent[2,4,17], we used antibiotic treatment to test whether CD4+CD8αα+ IEL abundance in *Ccr9−/−* mice was also microbiota-dependent. As seen in *Ccr9+/+* mice, antibiotic-treated *Ccr9−/−* mice resulted in a decreased frequency of CD4+CD8αα+ IELs compared with non-treated *Ccr9−/−* mice (Supplementary Fig. 3d–f). CCR9 expression in IELs remained unaffected by antibiotic treatment, suggesting that microbes did not modulate the abundance of CD4+CD8αα+ IELs through CCR9 expression (Supplementary Fig. 3g). Furthermore, the gut microbiota composition was found to be similar between *Ccr9+/+* and *Ccr9−/−* mice (Supplementary Fig. 3h, i). Fecal microbiota transfer from *Ccr9+/+* and *Ccr9−/−* mice to germ-free mice confirmed that the enhanced induction of CD4+CD8αα+ IELs in *Ccr9−/−* mice was not mediated by the altered microbiota composition (Supplementary Fig. 3j).

The differentiation of Treg cells and CD4+CD8αα+ IELs is reliant on RA[3,9,18–21]. Notably, vitamin A (VA) is indispensable for CCR9 expression, as it is involved in changes to chromatin accessibility[6,22]. To investigate the role of VA in inducing CD4+CD8αα+ IELs, we fed *Ccr9−/−* mice a VA-deficient diet. The frequency of CD4+CD8αα+ IELs in VA-deficient mice

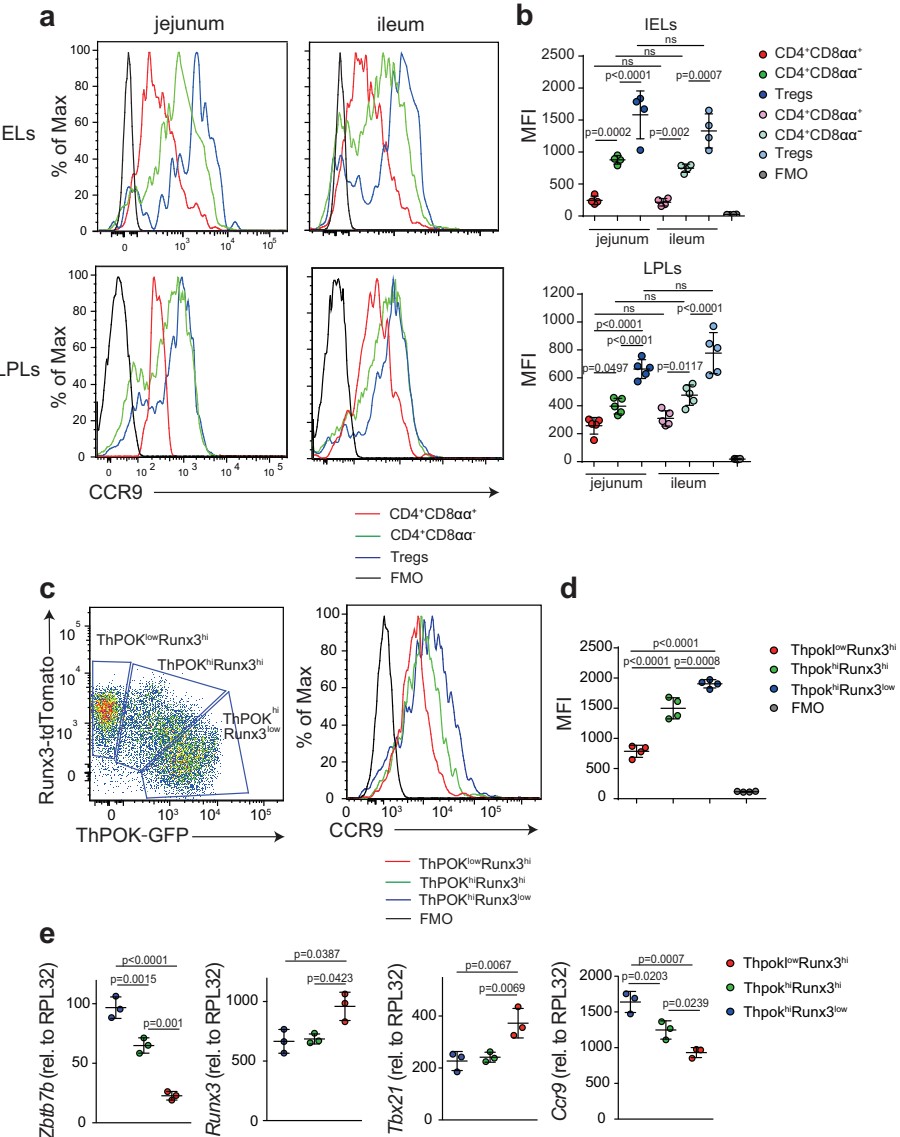

**Fig. 1 | CCR9 expression is lower in CD4⁺CD8αα⁺ IELs than in other IEL popu-lations. a** Histograms show the CCR9 level among TCRβ⁺CD4⁺CD8α⁺CD8β⁻ Foxp3⁻ (CD4⁺CD8αα⁺; red line), TCRβ⁺CD4⁺CD8α⁻CD8β⁻Foxp3⁻ (CD4⁺CD8αα⁻; green line), and TCRβ⁺CD4⁺CD8α⁻CD8β⁻ Foxp3⁺ (Treg cells; blue line) of IELs and LPLs in the jejunum and ileum. Fluorescence minus one (FMO) control is shown as black line. **b** Graphs show the mean fluorescence intensity (MFI) of CCR9 among CD4⁺CD8αα⁺, CD4⁺CD8αα⁻, Treg cells, and FMO control of IELs and LPLs in the jejunum and ileum (n = 4 C57BL/6J mice for IEL analysis, n = 5 C57BL/6J mice for LPL analysis, 10 weeks old). **c** Left; pseudocolor plot shows three populations (ThPOK^low Runx3^hi, ThPOK^hi Runx3^hi, and ThPOK^hi Runx3^low) among CD4⁺ SI IELs according to the expression of ThPOK and Runx3. Right; representative histogram shows the expression of CCR9 among ThPOK^low Runx3^hi (red line), ThPOK^hi Runx3^hi (green line), and ThPOK^hi Runx3^low (blue line) CD4⁺ SI IELs. FMO control is shown as black line. **d** Graph shows the MFI of CCR9 among ThPOK^low Runx3^hi, ThPOK^hi Runx3^hi, ThPOK^hi Runx3^low and FMO control of CD4⁺ SI IELs (n = 4 *Thpok*^GFP:*Runx3*^tdTomato reporter mice, 10 weeks old). **e** Graphs show relative expression of *Zbtb7b, Runx3, Tbx21,* and *Ccr9* in ThPOK^low Runx3^hi, ThPOK^hi Runx3^hi, and ThPOK^hi Runx3^low popu-lations among CD4⁺ SI IELs sorted from *Thpok*^GFP:*Runx3*^tdTomato reporter mice. Quantitative real-time PCR experiments were performed in duplicate in each sample, and each dots represented as average of duplicate (n = 3). Data are pre-sented as mean ± SD. One-way ANOVA with Tukey's multiple comparisons post hoc test was applied. Source data are provided as a Source data file.

was lower than that in the control group. These data suggested that RA was essential for CD4⁺CD8αα⁺ IEL differentiation in *Ccr9*^+/+ and *Ccr9*^−/− mice (Supplementary Fig. 3k).

We next examined whether naive CD4⁺ T cells from *Ccr9*^−/− mice had a higher likelihood of differentiating into CD4⁺CD8αα⁺ IELs. First, naive CD4⁺ T cells isolated from *Ccr9*^+/+ and *Ccr9*^−/− mice were cultured with anti-CD3/28, TGF-β, RA, 2,3,7,8-tetra-chlododibenzodioxin (TCDD), and IFN-γ (IEL differentiation factor) (Fig. 2d)[9,11]. CD4⁺CD8αα⁺ T cells were preferentially induced from the cells of *Ccr9*^−/− over those of *Ccr9*^+/+ mice (Fig. 2e, f and Supple-mentary Fig. 3l, m). The cultured T cells from *Ccr9*^−/− mice exhibited higher *Runx3* and *Tbx21* expression than those from *Ccr9*^+/+ mice

(Supplementary Fig. 3n). The expression of *Zbtb7b* in cultured T cells was comparable between *Ccr9*^+/+ and *Ccr9*^−/− mice (Supple-mentary Fig. 3n). We also attempted to culture naive CD4⁺ T cells from both mouse strains with CCL25 under IEL-differentiating condition (Supplementary Fig. 3l). However, CCL25 did not affect the percentage of CD4⁺CD8αα⁺ T cells nor the expression of *Runx3, Zbtb7b,* or *Tbx21* in *Ccr9*^+/+ and *Ccr9*^−/− mice (Supplementary Fig. 3m, n). These data suggest that *Ccr9*^−/− CD4⁺ T cells preferentially dif-ferentiated into CD4⁺CD8αα⁺ T cells in a CCL25-independent man-ner via increased expression of *Runx3* and *Tbx21*.

Next, we performed a bone marrow chimera experiment to investigate whether the increased CD4⁺CD8αα⁺ T cell abundance in

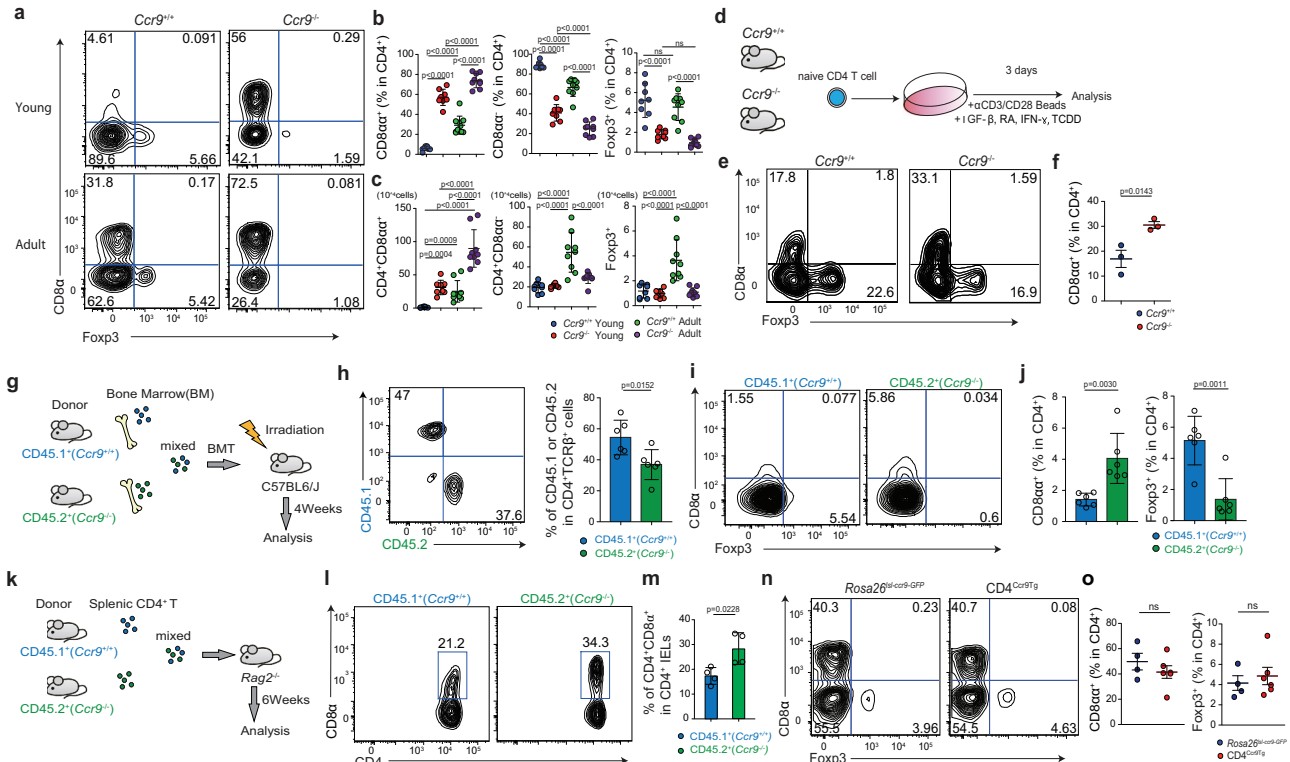

**Fig. 2 | CD4⁺CD8αα⁺ IELs are accumulated in *Ccr9⁻/⁻* mice, but surface CCR9 is dispensable for the induction of CD4⁺CD8αα⁺ IELs. a** Surface CD8α and intracellular Foxp3 expression by TCRβ⁺CD4⁺CD8β⁻ cells in SI IELs of *Ccr9⁺/⁺* and *Ccr9⁻/⁻* mice (Young; analyzed at 7 weeks old, Adult; analyzed at 10 weeks old). **b** Graphs show the frequency of CD8α⁺, CD8α⁻ or Foxp3⁺ population among TCRβ⁺CD4⁺CD8β⁻ cells in SI IELs of *Ccr9⁺/⁺* and *Ccr9⁻/⁻* mice. Data of young and adult mice were shown. Data are presented as mean ± SD. **c** Graphs show the total cell number of CD8α⁺, CD8α⁻ or Foxp3⁺ population among TCRβ⁺CD4⁺CD8β⁻ cells in SI IELs of *Ccr9⁺/⁺* and *Ccr9⁻/⁻* mice. Data of young and adult mice were shown. Data are presented as mean ± SD (**b, c**: *n* = 8 mice for young mice group, *n* = 9 mice for adult mice group). **d** Scheme of in vitro experiment design. Naïve CD4⁺ T cells obtained from *Ccr9⁺/⁺* and *Ccr9⁻/⁻* mice were cultured with anti-CD3/CD28, transforming growth factor-β (TGF-β), retinoic acid (RA), IFN-γ, and 2,3,7,8-Tetra-chlododibenzodioxin (TCDD). Surface CD8α and intracellular Foxp3 expression were analyzed after culture. **e** Surface CD8α and intracellular Foxp3 expression among TCRβ⁺CD4⁺CD8β⁻ cultured cells of *Ccr9⁺/⁺* and *Ccr9⁻/⁻* mice. **f** Graph shows the frequency of CD8α⁺ population among TCRβ⁺CD4⁺CD8β⁻ cultured cells of *Ccr9⁺/⁺* and *Ccr9⁻/⁻* mice. Data are presented as mean ± SEM (**d–f**: three independent experiments were performed in triplicate. Each dot represents the mean of the triplicate). **g** Schema of experiment design. Cells were obtained from bone marrow of *Cd45.1⁺Ccr9⁺/⁺* mice and *Cd45.2⁺ Ccr9⁻/⁻* mice and mixed 1:1 ratio. Mixed bone marrow cells were transferred to the lethally irradiated (11 Gy) C57BL/6J host mice

and mice were analyzed 4 weeks after transfer (*n* = 6 mice). **h** Surface CD45.1 and CD45.2 expression among TCRβ⁺CD4⁺CD8β⁻ cells in SI IELs. Graph shows the percentage of CD45.1⁺ and CD45.2⁺ cells in TCRβ⁺CD4⁺CD8β⁻ SI IELs. Data are presented as mean ± SD. **i** Surface staining of CD8α and intracellular Foxp3 among CD45.1⁺TCRβ⁺CD4⁺CD8β⁻ or CD45.2⁺TCRβ⁺CD4⁺CD8β⁻ cells in SI IELs. **j** Graphs show the percentage of CD8α and intracellular Foxp3 among CD45.1⁺TCRβ⁺CD4⁺CD8β⁻ or CD45.2⁺TCRβ⁺CD4⁺CD8β⁻ cells in SI IELs. Data are presented as mean ± SD. **k** Schema of experiment design. TCRβ⁺CD4⁺CD8β⁻ cells obtained from spleen of *Cd45.1⁺Ccr9⁺/⁺* mice and *Cd45.2⁺ Ccr9⁻/⁻* and mixed 1:1 ratio. Mixed cells were transferred to *Rag2⁻/⁻* mice and mice were analyzed 6 weeks after transfer (*n* = 4 mice). **l** Surface staining of CD8α among CD45.1⁺TCRβ⁺CD4⁺CD8β⁻ or CD45.2⁺TCRβ⁺CD4⁺CD8β⁻ cells in SI IELs. **m** Graph shows the percentage of CD8α among CD45.1⁺TCRβ⁺CD4⁺CD8β⁻ or CD45.2⁺TCRβ⁺CD4⁺CD8β⁻ cells in SI IELs. Data are presented as mean ± SD. **n** Surface CD8α and intracellular Foxp3 expression by TCRβ⁺CD4⁺CD8β⁻ cells in SI IELs of *Rosa26*ˡˢˡ⁻ᶜᶜʳ⁹⁻ᴳᶠᴾ and CD4ᶜᶜʳ⁹ᵀᵍ mice. **o** Graphs show the frequency of CD8α⁺ or Foxp3⁺ population among TCRβ⁺CD4⁺CD8β⁻ cells in SI IELs of *Rosa26*ˡˢˡ⁻ᶜᶜʳ⁹⁻ᴳᶠᴾ and CD4ᶜᶜʳ⁹ᵀᵍ mice (*n* = 4 mice for *Rosa26*ˡˢˡ⁻ᶜᶜʳ⁹⁻ᴳᶠᴾ group, *n* = 6 mice for CD4ᶜᶜʳ⁹ᵀᵍ group, 10 weeks old). Data expressed as mean ± SEM. One-way ANOVA with Tukey's multiple comparisons post hoc test (**b, c**) or the two-sided Student's *t* test (**f, h, j, m, o**) was applied. Source data are provided as a Source data file.

*Ccr9⁻/⁻* mice was cell autonomous. We transferred bone marrow cells from *Cd45.2⁺Ccr9⁻/⁻* and *Cd45.1⁺Ccr9⁺/⁺* mice to lethally irradiated C57BL/6J host mice, and the frequency of CD4⁺CD8αα⁺ IELs from each donor population was analyzed (Fig. 2g). The percentages of *Cd45.2⁺Ccr9⁻/⁻* derived CD4⁺TCRβ⁺ T cells was lower compared to *Cd45.1⁺Ccr9⁺/⁺*-derived CD4⁺TCRβ⁺ T cells in these irradiated mice in splenocytes, MLN, and IE compartment (Fig. 2h and Supplementary Fig. 3o). Notably, the frequency of CD4⁺CD8αα⁺ IELs differentiated from *Cd45.2⁺Ccr9⁻/⁻* cells was significantly higher than that from *Cd45.1⁺Ccr9⁺/⁺* cells (Fig. 2i, j).

Next, we transfer the same number of splenic CD4⁺ T cells from *Cd45.2⁺Ccr9⁻/⁻* and *Cd45.1⁺Ccr9⁺/⁺* mice into *Rag2⁻/⁻* mice. The percentage of CD4⁺CD8αα⁺ IELs derived from *Cd45.2⁺Ccr9⁻/⁻* cells was significantly higher than that derived from *Cd45.1⁺Ccr9⁺/⁺* cells (Fig. 2k–m).

Taken together, these results suggest that CCR9-deficient CD4⁺ T cells were intrinsically predisposed to develop into CD4⁺CD8αα⁺ IELs.

Finally, to further examine whether constitutive CCR9 expression influenced CD4⁺CD8αα⁺ IEL differentiation, we generated a *Rosa26*ˡˢˡ⁻ᶜᶜʳ⁹⁻ᴳᶠᴾ mouse strain and crossed it with the *Cd4*ᶜʳᵉ strain (*cd4*ᶜʳᵉ:*Rosa26*ˡˢˡ⁻ᶜᶜʳ⁹⁻ᴳᶠᴾ, hereafter referred to as CD4ᶜᶜʳ⁹ᵀᵍ) (Supplementary Fig. 4a, b). The frequency of CD4⁺CD8αα⁺ and Foxp3⁺ IELs was similar between *Rosa26*ˡˢˡ⁻ᶜᶜʳ⁹⁻ᴳᶠᴾ and CD4ᶜᶜʳ⁹ᵀᵍ mice (Fig. 2n, o), and the expression of CCR9 was similar between the CD4⁺CD8αα⁺ and CD4⁺CD8αα⁻ T cells of CD4ᶜᶜʳ⁹ᵀᵍ mice (Supplementary Fig. 4c), indicating that CCR9 overexpression did not prevent CD4⁺CD8αα⁺ IEL development. Thus, although *Ccr9⁻/⁻* CD4⁺ T cells strongly facilitated CD4⁺CD8αα⁺ IEL differentiation, CCR9 overexpression in CD4⁺ T cells was not sufficient to prevent this process.

## *Ccr9*[−/−] CD4[+] T cells facilitate the development of CD4[+]CD8αα[+] IELs via CD4[+]CD8αα[−] precursors

We next investigated whether *Ccr9*[−/−] CD4[+] T cells induced cell-intrinsic changes in peripheral CD4[+] T cells using droplet-based scRNA-seq in the Chromium 10X platform. We analyzed the gene expression profiles of CD4[+] T cells from spleen tissues, MLNs, LPLs, and IELs sorted from *Ccr9*[+/+] and *Ccr9*[−/−] mice (spleen, MLN: *n* = 1, LPLs, IELs: *n* = 3, cell number: Supplementary Fig. 5a). We identified 17 clusters based on the differentially expressed genes (DEGs), which were visualized by uniform manifold approximation and projection (UMAP) (Fig. 3a, b). We focused on clusters 0−14 because clusters 15 and 16 represented only a

small fraction of the cells. When cells were segregated by tissue type, the fraction of IELs was mainly represented by clusters 0, 1, 2, 3, 4, 7, 10, 13, and 14. Among these clusters, clusters 0, 4, and 7 expressed CD4[+]CD8αα[+] IEL-associated genes (*Cd8a*, *Ccl5*, *Gzma, Nkg7*, and *Itgae*) and were predominantly derived from *Ccr9*[−/−] mice (Fig. 3c–f, Supplementary Fig. 5b, c, and Supplementary Data 1), reflecting the increased number of CD4[+]CD8αα[+] IELs in this strain. Clusters 10, 13, and 14 also expressed CD4[+]CD8αα[+] IEL-associated genes, but with slightly reduced expression of *Cd8a* compared to that in clusters 0, 4, and 7. Thus, clusters 10, 13, and 14 were considered CD4[+]CD8α[int] IELs (Supplementary Fig. 5d and Supplementary Data 2).

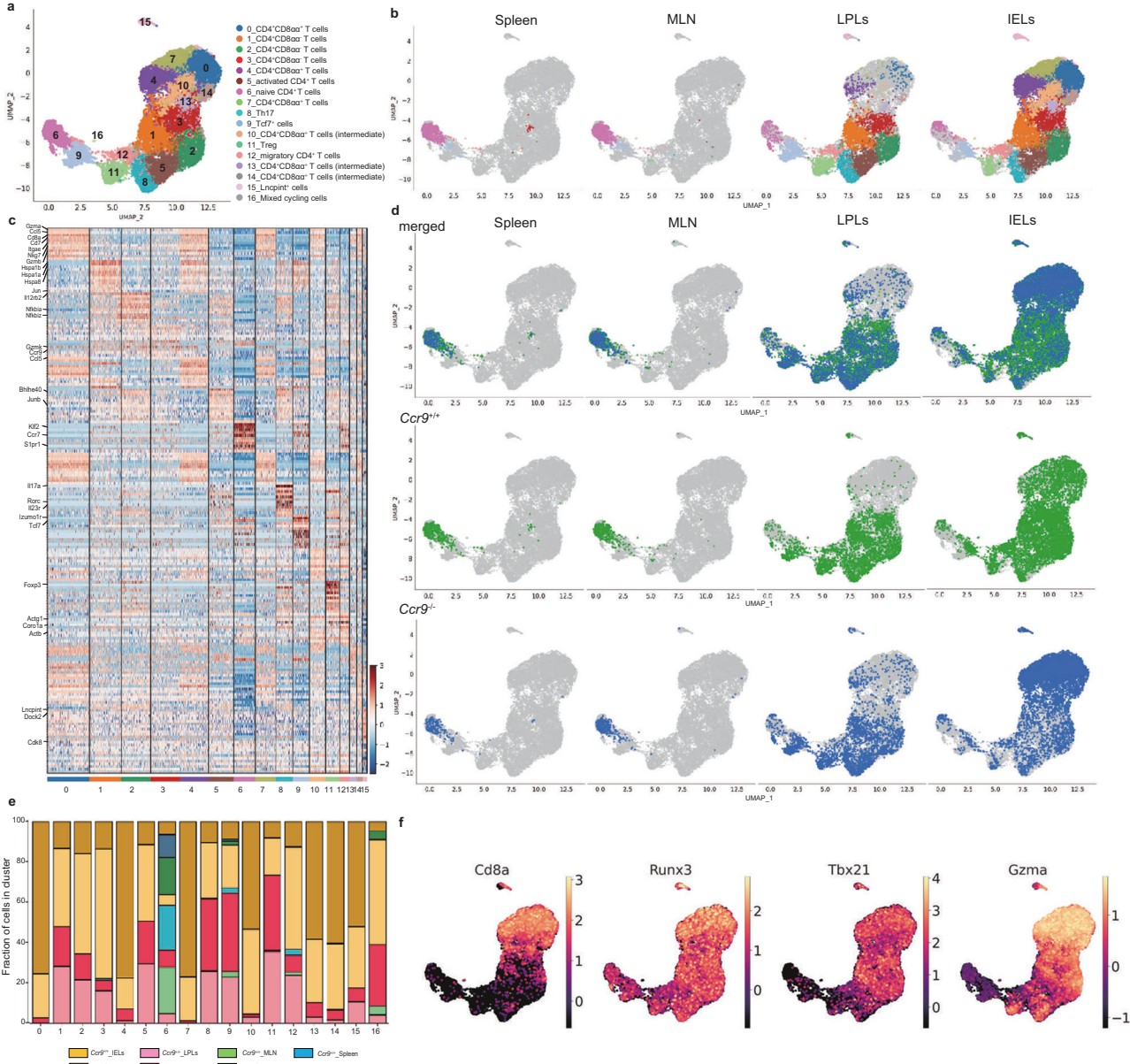

**Fig. 3 | CD4[+] T cells in the intraepithelial compartment are differentially distributed between *Ccr9*[+/+] and *Ccr9*[−/−] mice.** Droplet-based single-cell RNA sequencing (scRNA-seq) was performed using the Chromium 10X platform. CD4[+] T cells from the spleen, mesenteric lymph nodes (MLNs), LPLs and IELs were sorted from *Ccr9*[+/+] and *Ccr9*[−/−] mice for scRNA-seq. CD4[+] T cells from spleen and MLNs were collected from one *Ccr9*[+/+] and *Ccr9*[−/−] mice, and LPLs and IELs were collected from three *Ccr9*[+/+] and three *Ccr9*[−/−] mice. Sorted IELs and LPLs were pooled in a 2:1 ratio, and cells of each tissue were pooled in a 1:1 ratio between *Ccr9*[+/+] and *Ccr9*[−/−] mice. **a** Cells were positioned by gene expression similarities, and 17 clusters were

identified based on their top differentially expressed genes (DEGs), as visualized by uniform manifold approximation and projection (UMAP). **b** UMAP representation of all sequenced cells, color-coded by cluster and separated by tissue of origin. **c** Heatmap of the top DEGs in each cluster. **d** UMAP representation of all sequenced cells and separated by tissue of origin. Cell distribution in each tissue from *Ccr9*[+/+] mice (middle), *Ccr9*[−/−] mice (bottom), and both (top) is shown. **e** Bar graph shows the proportion of cells in each cluster originating from the spleen, MLN, LPLs, and IELs of *Ccr9*[+/+] and *Ccr9*[−/−] mice. **f** Expression of *Cd8a*, *Runx3*, *Tbx21* and *Gzma* in sequenced cells are shown via UMAP. Source data are provided as a Source data file.

It was observed that cells in clusters 1, 2, and 3 lacked *Cd8a* expression, representing CD4+CD8αα− IEL populations (Fig. 3c, f). Cluster 1 expressed *Hspa* and *Jun*, whereas cluster 2 expressed *Il12rb* and included Th1 cells. Cluster 3 displayed a cytotoxic profile (*Gzmk* and *Ccl5*) and high expression of *Ccr9* (Fig. 3c and Supplementary Fig. 5b, c).

It was observed that cluster 6 primarily comprised naive cell markers (*Sell*, *S1pr1*, *Klf2*, and *Ccr7*) and contained cells mainly from spleen and MLN tissues (Fig. 3b–e and Supplementary Fig. 5b, c). Conversely, clusters 5, 8, 9, and 11 consisted mainly of LPL cells and exhibited the expression of different types of genes in activated T cells, such as *Bhlhe40* in cluster 5; *Il17a*, *Il22*, *Il23r*, and *Rorc* in cluster 8 (Th17); self renewal associated genes, such as *Tcf7* and *Izumo1r* in cluster 9 (Tcf7 cluster); and *Foxp3* in cluster 11 (Treg cells) (Fig. 3c and Supplementary Fig. 5b, c). Additionally, cluster 12 contained IELs and LPLs expressing motility-associated genes such as *Coro1a*, *Actb*, and *Actg1*. (Fig. 3c, e and Supplementary Fig. 5b, c).

Given that we included almost the same number of CD4+ T cells between *Ccr9+/+* and *Ccr9−/−* mice (Supplementary Fig. 5a), we focused on the cell distributions between *Ccr9+/+* and *Ccr9−/−* mice (each mice data, Supplementary Fig. 6a, b). CD4+ T cells in cluster 6 from the spleen, MLN, and LPL tissues displayed the same clustering distributions between *Ccr9+/+* and *Ccr9−/−* mice. CD4+CD8αα+ T cells in clusters 0, 4, and 7 primarily came from *Ccr9−/−* mice (Fig. 3d, e). CD4+CD8αα− T cells in cluster 3 were mainly IELs in *Ccr9+/+* mice (Fig. 3d, e). Notably, clusters 0, 3, 4, and 7 predominantly comprised IELs. These data suggested that the distribution of cells differed between *Ccr9+/+* and *Ccr9−/−* mice, especially regarding CD4+CD8αα− IELs and CD4+CD8αα+ IELs.

Clusters 1, 2, and 3 were considered to represent CD4+CD8αα+ IEL "progenitors". Trajectory analysis revealed that these CD4+CD8αα+ IEL progenitors preceded clusters 7 in *Ccr9+/+* mice, but preceded the development of clusters 4, and 7 in *Ccr9−/−* mice (Fig. 4a). Pseudotime analysis further revealed that some CD4+ T cells from *Ccr9+/+* mice stopped at clusters 1, 2, and 3, whereas CD4+ T cells from *Ccr9−/−* mice passed through these clusters and differentiated into CD4+CD8αα+ IELs. These data suggest that CD4+ T cells from *Ccr9+/+* mice and *Ccr9−/−* mice may develop differently in IE compartment, resulting in low frequency in cluster 3 from *Ccr9−/−* mice and high frequency in clusters 0, 4, and 7.

The differentiation and expansion of CD4+CD8αα+ IELs with low diversity is influenced by T cell receptor (TCR) signaling, which may regulate the size and heterogeneity of these cells[23,24]. To investigate whether specific TCR selection affects CD4+CD8αα+ IELs in *Ccr9−/−* mice, we examined TCR repertoire diversity in various tissues. The Morisita–Horn index indicated that IELs were similar to LPLs both in *Ccr9+/+* and *Ccr9−/−* mice (Supplementary Fig. 7a). TCR repertoire diversity was also similar among CD4+ T cells from spleen, MLN, and LPL tissues between *Ccr9+/+* and *Ccr9−/−* mice (Supplementary Fig. 7b). However, the clonality of TCR in IELs from *Ccr9−/−* mice was higher than that in *Ccr9+/+* mice (Supplementary Fig. 7b). Additionally, the top 10 clonotypes corresponded to approximately 50% of all clonotypes in *Ccr9−/−* mice (Supplementary Fig. 7c, each mouse data in Supplementary Fig. 7d). Therefore, TCR diversity in the spleen, MLN, and LPLs was comparable between *Ccr9+/+* and *Ccr9−/−* mice, but TCR diversity in IELs from *Ccr9−/−* mice was reduced compared with that in *Ccr9+/+* mice. These data indicate that specific TCR selection did not occur in *Ccr9−/−* mice until they migrated to the LP, and that T cell heterogeneity was reduced in IELs.

We then investigated possible differences in the distribution of clonally expanded cells between *Ccr9+/+* and *Ccr9−/−* mice. We selected the top five clonotypes in IELs from *Ccr9+/+* and *Ccr9−/−* mice (Supplementary Fig. 7e). While these clonotypes were observed both in CD4+CD8αα+ IEL precursors (clusters 1, 2, and 3) and CD4+CD8αα+ IELs in *Ccr9+/+* mice, most were not observed in CD4+CD8αα+ IEL precursors but rather in CD4+CD8αα+ IELs (including cluster 0, 4, and 7) in *Ccr9−/−*

mice (Supplementary Fig. 7e). Shannon's index in cluster 0, 4, and 7 in IELs from *Ccr9−/−* mice was lower than that from *Ccr9+/+* mice (Supplementary Fig. 7f), suggesting that CD4+CD8αα+ IELs of *Ccr9−/−* mice clonally expanded. Furthermore, the expression of cell growth markers EdU and Ki67 in CD4+CD8αα+ IELs in *Ccr9−/−* mice was higher than in CD4+CD8αα+ IELs in *Ccr9+/+* mice, but was not different significantly between CD4+CD8αα− IELs in *Ccr9+/+* and *Ccr9−/−* mice (Supplementary Fig. 7g). Taken together, these results suggest that CD4+CD8αα+ IELs in *Ccr9−/−* mice represented a more proliferative cell population than those in *Ccr9+/+* mice.

Because we observed CD4+CD8αα+ T cells both in the IE compartment and the LP of *Ccr9−/−* mice (Fig. 3d and Supplementary Fig. 3c), we examined whether CD4+CD8αα+ T cells in the IE compartment of *Ccr9−/−* mice were different from those in the LP of *Ccr9−/−* mice. CD4+CD8αα+ T cells in the IE compartment and the LP from *Ccr9−/−* mice expressed *Runx3*, *Cbfb*, and *CD8a* (Supplementary Fig. 7h). These data indicate that CD4+CD8αα+ T cells in the IE compartment and the LP of *Ccr9−/−* mice were similar gene expression.

## *Ccr9−/−* CD4+ T cells enhance the CD4+CD8αα+ cell program

Next, we analyzed the expression of genes encoding transcription factors known to drive the differentiation of CD4+CD8αα+ IELs, such as *Runx3*, *Tbx21* (encoding Tbet), and *Zbtb7b* (encoding ThPOK) in IEL clusters, CD4+CD8αα+ IEL progenitors clusters (cluster 1, 2, and 3), CD4+CD8αint IEL clusters (cluster 10, 13, and 14), and CD4+CD8αα+ IEL clusters (clusters 0, 4, and 7) (Fig. 4b, Supplementary Fig. 7i, and Supplementary Data 2). The expression of both *Runx3* and *Tbx21* in the CD4+CD8αα+ IEL cluster was higher than in the CD4+CD8αα+ IEL progenitor and CD4+CD8αint IEL clusters. Further analysis revealed that the expression of *Runx3* and *Tbx21* in CD4+CD8αα+ IEL progenitors, CD4+CD8αint IELs (cluster 10), and CD4+CD8αα+ IELs of *Ccr9−/−* mice was higher than that in all clusters of *Ccr9+/+* mice (Fig. 4c and Supplementary Fig. 7j). Therefore, high expression of CD4+CD8αα+ IEL–related genes in *Ccr9−/−* mice promoted CD4+CD8αα+ IEL development, resulting in the accumulation of CD4+CD8αα+ IELs in *Ccr9−/−* mice.

Interestingly, we noticed a higher level of *Cbfb*, which encodes an essential partner protein of RUNX3, in CD4+CD8αα+ IEL clusters (clusters 0,4 and 7) compared with CD4+CD8αα+ IEL progenitors clusters (clusters 1, 2 and 3) (Fig. 4b and Supplementary Data 2). Furthermore, *Cbfb* expression was higher in the CD4+CD8αα+ IEL progenitor clusters (clusters 1, 2, and 3), CD4+CD8αint IEL clusters (clusters 10 and 14), and CD4+CD8αα+ IEL clusters (clusters 0,4 and 7) of *Ccr9−/−* mice than in the same clusters of *Ccr9+/+* mice (Fig. 4c). Then, we investigated whether the expression of *Cbfb1*, *Cbfb2*, *Runx3*, and *Tbx21* in naive CD4+ T cells from *Ccr9−/−* mice was higher than in *Ccr9+/+* mice. Quantitative real-time PCR analysis revealed that *Cbfb1* and *Cbfb2* expression was similar between naive CD4+ T cells from *Ccr9+/+* and *Ccr9−/−* mice (Fig. 5a). Although *Cbfb1* levels in CD4+CD8αα− and CD4+CD8αα+ IELs were similar between *Ccr9+/+* and *Ccr9−/−* mice, *Cbfb2* levels in these two cell populations were higher in *Ccr9−/−* mice than in *Ccr9+/+* mice (Fig. 5a). *Runx3* and *Tbx21* mRNA expression in CD4+CD8αα− and CD4+CD8αα+ IELs was higher in *Ccr9−/−* mice than in *Ccr9+/+* mice (Fig. 5a). *Zbtb7b* expression was similar between *Ccr9+/+* and *Ccr9−/−* mice (Supplementary Fig. 7k). These results showed that the increased expression of *Cbfb2*, *Runx3*, and *Tbx21* of CD4+ T cells in *Ccr9−/−* mice relative to that in *Ccr9+/+* mice did not begin in naive CD4+ T cells but in CD4+CD8αα− and CD4+CD8αα+ IELs.

We further analyzed *Cbfb1* and *Cbfb2* expression in the three IEL subsets distinguished by *Thpok*GFP:*Runx3*tdTomato reporter alleles. *Cbfb2* expression was higher in ThPOKlowRunx3hi cells than in ThPOKhiRunx3hi and ThPOKhiRunx3low cells, whereas *Cbfb1* expression was consistent across all three subsets (Fig. 5b). These data were further supported by experiments using *Cbfb1*tdTomato and *Cbfb2*Venus reporter alleles to reflect *Cbfb1* and *Cbfb2* splicing by expression of tdTomato and Venus

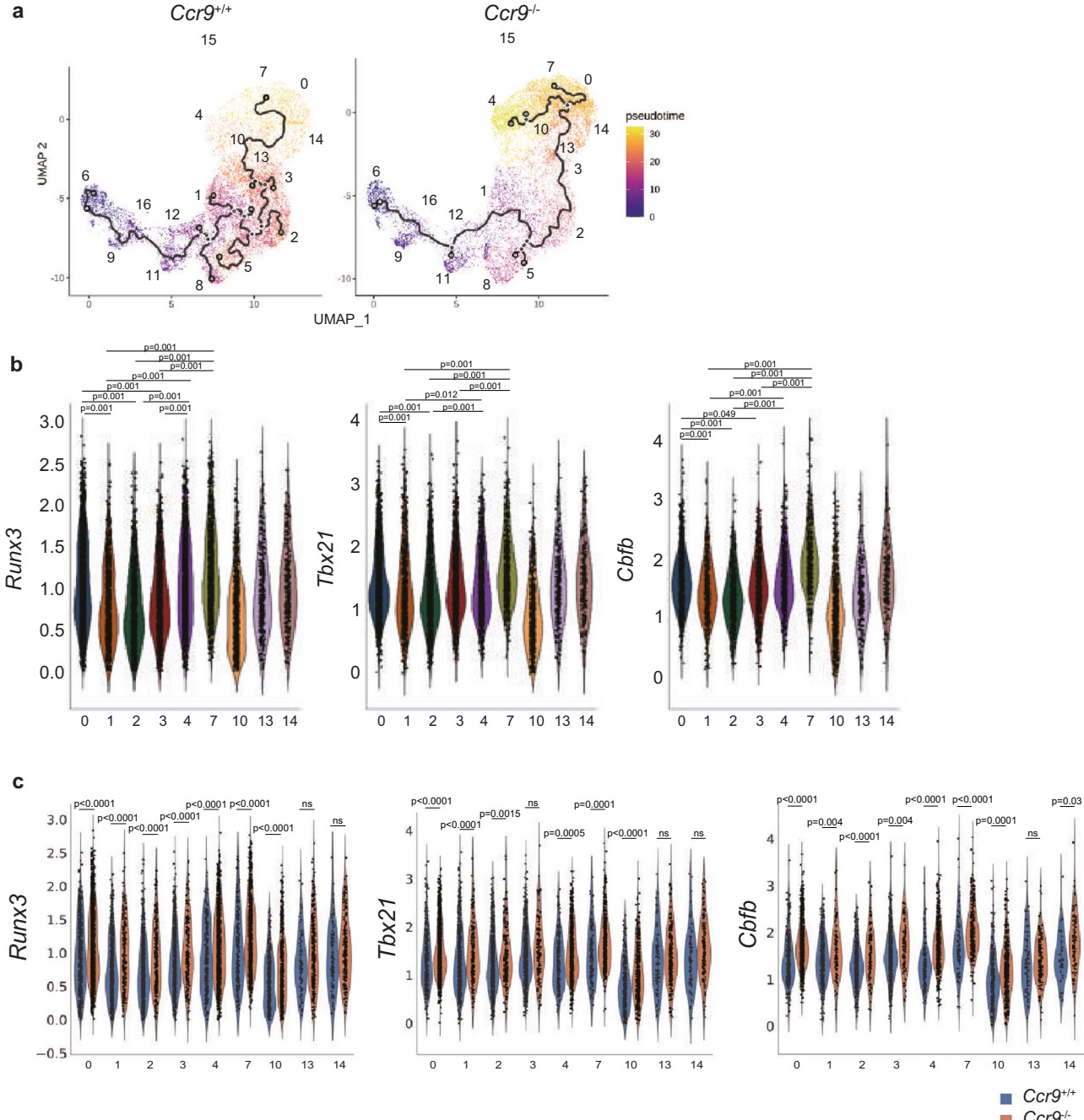

**Fig. 4 | CD4+CD8αα+ IEL-related genes are abundant in Ccr9−/− mice.**
**a** Pseudotime analysis of sequenced IELs originating from Ccr9+/+ mice (left) and Ccr9−/− mice (right), color-coded by pseudotime gradient. Numbers on the UMAP figure indicate the cluster numbers. **b** Violin plots show expression of Runx3, Tbx21, and Cbfb among IELs from clusters 0, 4, and 7 (CD4+CD8αα+ T cells); clusters 1, 2 and 3 (CD4+CD8αα− T cells); and clusters 10, 13 and 14 (CD4+CD8αα^int T cells).

**c** Violin plots show the expression of Runx3, Tbx21, and Cbfb among IELs from clusters 0, 4, and 7 (CD4+CD8αα+ T cells); clusters 1, 2 and 3 (CD4+CD8αα− T cells); and clusters 10, 13, and 14 (CD4+CD8αα^int T cells) from Ccr9+/+ and Ccr9−/− mice (**a**−**c**; n = 3 biologically independent IEL samples for each strain). One-way ANOVA with Tukey's multiple comparisons post-hoc test (**b**) or the two-sided Student's t test (**c**) was applied. Source data are provided as a Source data file.

fluorescent proteins, respectively. Cbfβ2-Venus expression was upregulated in CD4+CD8αα+ IELs, whereas Cbfβ1-tdTomato expression was similar between CD4+CD8αα− and CD4+CD8αα+ IELs (Fig. 5c). Our findings suggest that Cbfβ2 but not Cbfβ1 may have been crucial for the development of CD4+CD8αα+ IELs.

**Cbfβ2 expression regulates CD4+CD8αα+ IEL differentiation**
Upregulation of Cbfβ2 expression during CD4+CD8αα+ IEL differentiation prompted us to examine whether Cbfβ2 regulates CD4+CD8αα+ IEL differentiation using a Cbfb2m/2m mouse strain,

in which mutation of the splicing-donor sequence for Cbfb2 abrogated Cbfβ2 production. The results showed that the lack of Cbfβ2 reduced the population of CD4+CD8αα+ IELs but did not affect the composition of Treg cells or effector cells among LPLs and MLNs. (Fig. 6a, b and Supplementary Fig. 8a−f). We next investigated whether continuous overexpression of Cbfβ2 promoted the development of CD4+CD8αα+ IELs. We generated Rosa26^lsl-Cbfb2-GFP mice that continuously expressed Cbfβ2 upon Cre-mediated excision of translational stop sequences. Crossing Cd4^cre mice with Rosa26^lsl-Cbfb2-GFP mice (Cd4^cre:Rosa26^lsl-Cbfb2-GFP; CD4^Cbfb2Tg) resulted in continuous expression of Cbfβ2 in T cells from

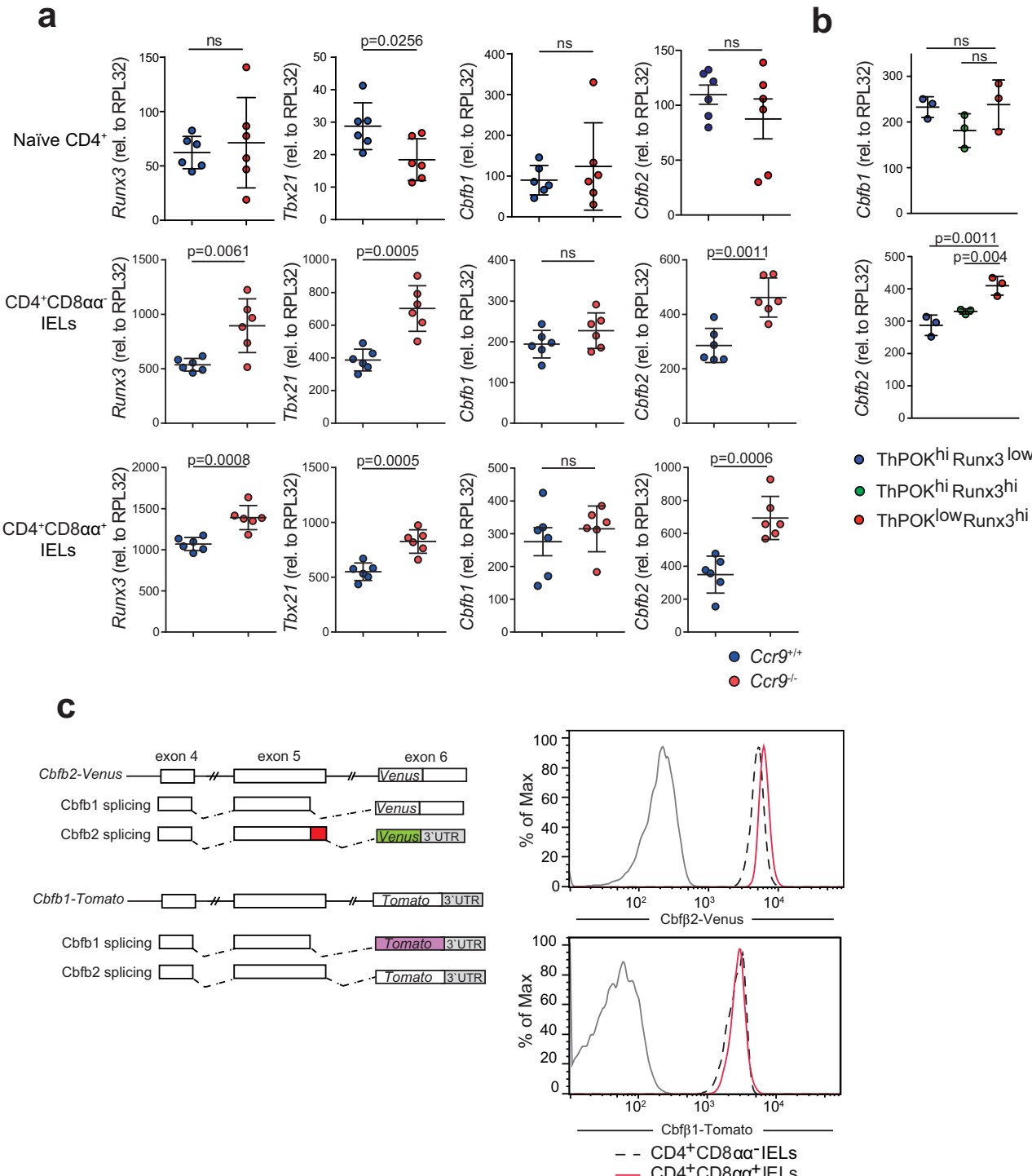

**Fig. 5 | *Cbfb2* expression is upregulated in CD4⁺CD8αα⁺ IELs of *Ccr9*⁻/⁻ mice.**
**a** Graphs show the relative expression of *Runx3*, *Tbx21*, *Cbfb1*, and *Cbfb2* in splenic naïve CD4⁺ T cells and in CD4⁺CD8αα⁻ and CD4⁺CD8αα⁺ IELs from *Ccr9*⁺/⁺ and *Ccr9*⁻/⁻ mice. Quantitative real-time PCR experiments were performed in duplicate in each sample, and each dots represented as average of duplicate (*n* = 6). **b** Graphs show the relative expression of *Cbfb1* and *Cbfb2* in ThPOKʰⁱRunx3ˡᵒʷ, Runx3ʰⁱThPOKʰⁱ, and ThPOKˡᵒʷRunx3ʰⁱ populations among CD4⁺ SI IELs sorted from *Thpok*ᴳᶠᴾ:*Runx3*ᵗᵈᵀᵒᵐᵃᵗᵒ reporter mice. Quantitative real-time PCR experiments were performed in duplicate in each sample, and each dots represented as average of duplicate (*n* = 3). **c** Left: Structure of a mouse *Cbfb* locus that produces two splice

variants by distinct splice donor signals within exon 5. cDNA encoding tdTomato or Venus was targeted into exon 6 in the *Cbfb* locus to monitor *Cbfb1* and *Cbfb2* splicing by expression of tdTomato and Venus fluorescent protein, respectively. Right: histograms show Cbfb2-Venus and Cbfb1-tdTomato expression in CD4⁺CD8αα⁻ IELs (dotted line) and CD4⁺CD8αα⁺ IELs (red line) from *Cbfb2*ᵛᵉⁿᵘˢ and *Cbfb1*ᵗᵈᵀᵒᵐᵃᵗᵒ reporter alleles, as well as CD4⁺CD8αα⁺ IELs (black line) from wild-type control mice. Data are expressed as mean ± SD. The two-sided Student's *t* test (**a**) or one-way ANOVA with Tukey's multiple comparisons post hoc test (**b**) was applied. Source data are provided as a Source data file.

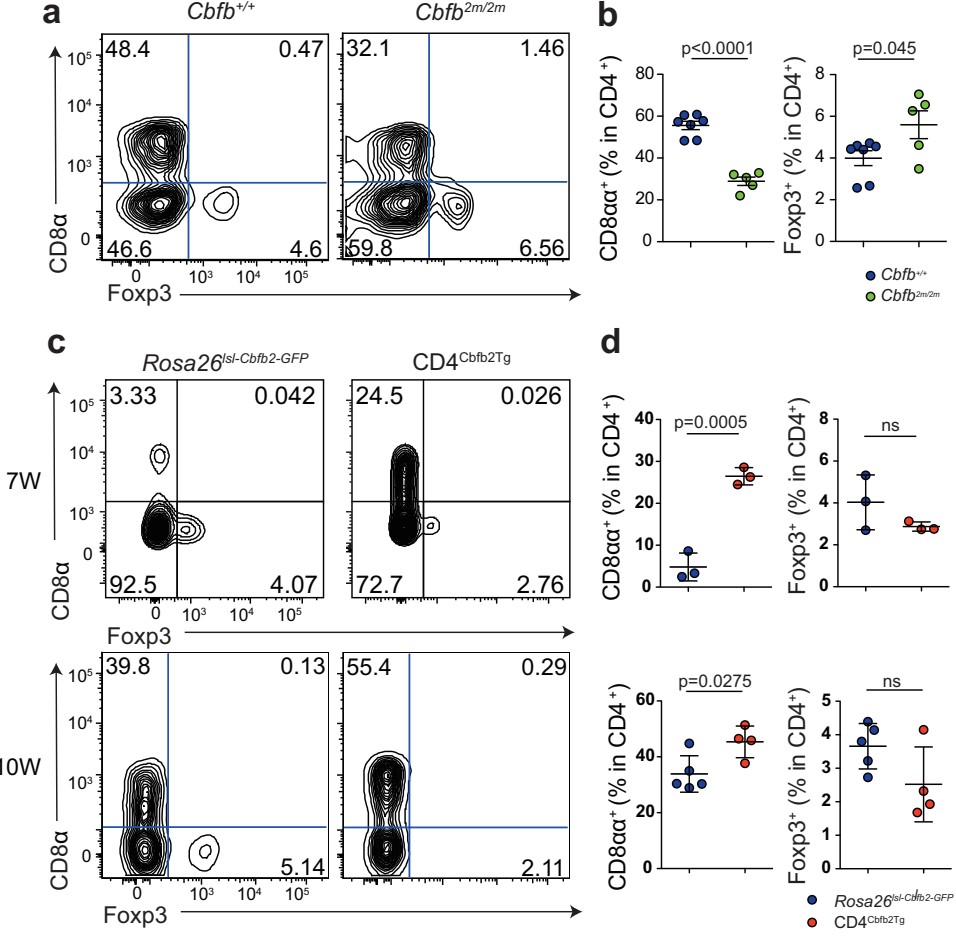

**Fig. 6 | Cbfβ2 expression regulates the induction of CD4⁺CD8αα⁺ IELs. a** Surface CD8α and intracellular Foxp3 expression of TCRβ⁺CD4⁺CD8β⁻ SI IELs from *Cbfb*⁺/⁺ and *Cbfb*²ᵐ/²ᵐ mice. **b** Graphs show the frequency of CD8α⁺ or Foxp3⁺ populations among TCRβ⁺CD4⁺CD8β⁻ cells in SI IELs. (*n* = 7 mice for *Cbfb*⁺/⁺, *n* = 5 mice for *Cbfb*²ᵐ/²ᵐ group, 10 weeks old). Data are expressed as mean ± SEM. **c** Surface CD8α and intracellular Foxp3 expression by TCRβ⁺CD4⁺CD8β⁻ cells in SI IELs of *Rosa26*ᴵˢˡ⁻ *Cbfb2*-GFP and CD4Cbfb2Tg mice analyzed at 7 and 10 weeks of age. **d** Graphs show the frequency of CD8α⁺ or Foxp3⁺ populations among TCRβ⁺CD4⁺CD8β⁻ SI IELs of *Rosa26*ᴵˢˡ⁻*Cbfb2*-GFP and CD4Cbfb2Tg mice (7W; *n* = 3 mice for each group, 10W; *n* = 5 mice for *Rosa26*ᴵˢˡ⁻*Cbfb2*-GFP group, *n* = 4 mice for CD4Cbfb2Tg group). Data are expressed as mean ± SD. The two-sided Student's *t* test (**b**, **d**) was applied. Source data are provided as a Source data file.

the double-positive thymocyte stage onwards. The frequency of CD4⁺CD8αα⁺ IELs in CD4^Cbfb2Tg mice was higher than that in *Rosa26*ᴵˢˡ⁻*Cbfb2*-GFP mice; this was not observed in *Cbfb*²ᵐ/²ᵐ mice (Fig. 6c, d). The frequency of Treg cells among LPLs was higher in CD4^Cbfb2Tg mice than in *Rosa26*ᴵˢˡ⁻*Cbfb2*-GFP mice (Supplementary Fig. 8g, h). Although the frequency of IL-17A-expressing cells was similar between CD4^Cbfb2Tg and *Rosa26*ᴵˢˡ⁻*Cbfb2*-GFP mice, that of IFN-γ-expressing cells was lower in CD4^Cbfb2Tg mice (Supplementary Fig. 8i, j). Furthermore, the frequency of Treg cells in MLN tissue was similar between CD4^Cbfb2Tg and *Rosa26*ᴵˢˡ⁻ *Cbfb2*-GFP mice (Supplementary Fig. 8k, l). These data suggest that the expression of Cbfβ2 was linked to the percentage of CD4⁺CD8αα⁺ IELs and influenced the composition of Treg cells and effector T cells.

To explore how Cbfβ2 regulates CD4⁺CD8αα⁺ IEL differentiation, we next analyzed CBFβ2 binding sites in *Runx3* and *Tbx21* by chromatin immunoprecipitation sequencing (ChIP-seq) in splenic naive CD4⁺ T cells, CD4⁺CD8αα⁻ IELs, and CD4⁺CD8αα⁺ IELs. CBFβ2 bound to the distal and proximal promoter regions of the *Runx3* and the *Tbx21* transcription site of CD4⁺CD8αα⁻ and CD4⁺CD8αα⁺ IELs, but not of naive CD4⁺ T cells (Supplementary Fig. 9a). CBFβ2 binding motifs were similar between CD4⁺CD8αα⁻ and CD4⁺CD8αα⁺ IELs, but different in naive CD4⁺ T cells (Supplementary Fig. 9b). Cbfβ2-specific regions contained erythroblast transformation specific (ETS) recognition sites (Supplementary Fig. 9c). ETS transcription factors have been reported to increase the DNA-binding affinity of the RUNX/CBFβ complex[25].

These data suggest that CBFβ2 binding sites were altered during IEL development, and that RUNX3/CBFβ2 complexes enhanced *Runx3* expression and induced *Tbx21* expression. Collectively, these results showed that upregulation of Cbfβ2 in CD4⁺ T cells facilitated their differentiation towards CD4⁺CD8αα⁺ T cells.

## Loss of CCR9 leads to cell-intrinsic changes during CD4⁺CD8αα⁺ IELs differentiation via *Cbfb*, *Tbx21*, and *Runx3*

Because CD4⁺ T cells in *Ccr9*⁻/⁻ mice developed into CD4⁺CD8αα⁺ T cells in the IE compartment, we hypothesized these cells might use the same molecules as CD4⁺ T cells in *Ccr9*⁺/⁺ to differentiate into CD4⁺CD8αα⁺ T cells. To investigate the factors involved in the development of CD4⁺CD8αα⁺ T cells, we examined their induction under IEL-differentiating conditions (Fig. 7a). The inhibition of *Cbfb*, *Tbx21*, and *Runx3* by small interfering RNA (siRNA) reduced the induction of CD4⁺CD8αα⁺ T cells compared to that observed with scramble siRNA (Fig. 7b). Additionally, the downregulation of *Ccr9* via siRNA in naive CD4⁺ T cells promoted the induction of CD4⁺CD8αα⁺ T cells compared to that observed under scramble siRNA treatment. Thus, reduction of *Ccr9* from CD4⁺ T cells encouraged differentiation into CD4⁺CD8αα⁺ T cells under IEL-differentiating conditions. Further analysis of CD4⁺ T cells from *Ccr9*⁻/⁻ mice revealed that their high levels of CD4⁺CD8αα⁺ T cells were dependent on *Cbfb*, *Tbx21*, and *Runx3*

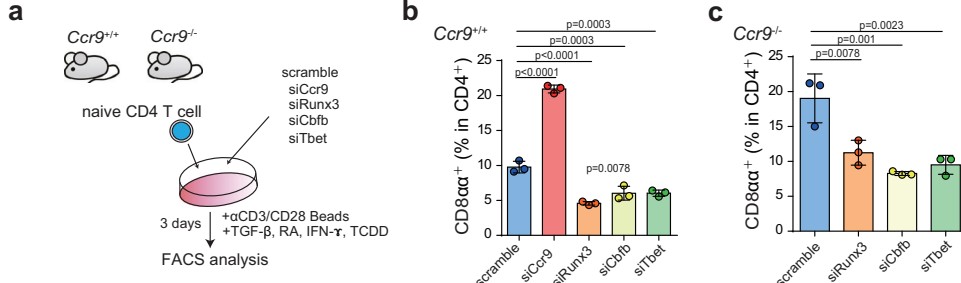

**Fig. 7 | Downregulation of *Ccr9* in CD4+ T cells preferentially develops CD4+CD8αα+ IELs in a *Cbfb*-dependent manner. a** Schema of the experimental design. Naive CD4+ T cells obtained from *Ccr9+/+* and *Ccr9−/−* mice were nucleofected with siRNA targeting negative control, *Ccr9, Runx3, Cbfb*, and *Tbx21* (scramble, siCcr9, siRunx3, siCbfb, and siTbet, respectively). Nucleofected cells were cultured with anti-CD3/CD28, transforming growth factor-β (TGF-β), retinoic acid (RA), 2,3,7,8-tetrachlodododibenzodioxin (TCDD), and IFN-γ. Surface CD8α and intracellular Foxp3 expression were analyzed 3 days after culture. **b, c** Graphs show the frequency of CD8α+ cells among TCRβ+CD4+CD8− nucleofected naive CD4+ T cells obtained from *Ccr9+/+* and *Ccr9−/−* mice cultured with anti-CD3/CD28, TGF-β, RA, TCDD, and IFN-γ. Three independent experiments were performed in triplicate. Each dot (*n* = 3) represents the mean of the triplicate. Data are expressed as mean ± SD. One-way ANOVA with Tukey's multiple comparisons post hoc test was applied. Source data are provided as a Source data file.

(Fig. 7c). Therefore, *Cbfb, Tbx21*, and *Runx3* played a crucial role in the induction of CD4+CD8αα+ T cells in *Ccr9−/−* mice.

## Discussion

In this study, we discovered that CD4+CD8αα+ IELs exhibited downregulated CCR9 corresponding to the induction of *Runx3* and *Tbx21* expression. Our results suggest that the loss of CCR9 facilitates CD4+ T cell precursors to acquire an IEL program, highlighting an important regulatory mechanism in the differentiation of CD4+CD8αα+ IELs.

The migration of pre-thymic T cell progenitors to the thymus is mediated by CCR7, CCR9, and P-selectin glycoprotein ligand-1 (PSGL1), which interact with their respective ligands, i.e., CCL19, CCL21, CCL25, and P-selectin, which are expressed by thymic endothelium cells[26–31]. Although CCR9 is temporarily downregulated after reaching the thymus, it is later re-expressed in response to RA in the MLNs and Peyer's patches, enabling gut tropic T cells to migrate to the small intestine via interaction with CCL25 secreted by small intestinal epithelial cells[6,7,32]. Thus, CCR9 is generally known to function as a regulator of αβTCR T cells homing at several developmental stages (i.e., migration to the thymus and small intestine). However, our research demonstrates that the development of CD4+CD8αα+ IELs is not affected by the absence of CCL25, indicating that migration to the small intestine is not involved in the development of CD4+CD8αα+ IELs. Indeed, the expression level of CCR9 in the IELs of *Ccl25−/−* mice was higher than that of *Ccl25+/+* mice, which might be due to the lack of CCL25-CCR9 interaction. Nevertheless, CCR9 expression was lower in CD4+CD8αα+ IELs in *Ccl25−/−* mice compared with other populations of Treg cells and CD4+CD8αα− IELs in *Ccl25−/−* mice. These data raised the possibility that the reduction in CCR9 expression, rather than the absolute expression of CCR9, may be involved in the development of CD4+CD8αα+ IELs.

While a lack of CCR9 expression led to an increased percentage of CD4+CD8αα+ IELs, overexpression of CCR9 did not prevent CD4+CD8αα+ IEL differentiation. We propose several possibilities to explain this phenomenon. First, epigenetic and chromatin modifications in CD4 T cells are essential during the development of IELs[5], and a lack of *Ccr9* might enhance these modifications, facilitating the adaptation of CD4+CD8αα+ IELs with increased expression of CD4+CD8αα+ IEL-related genes, such as *Runx3, Cbfb2*, and *Tbx21*. However, overexpression of CCR9 alone may not be adequate to induce the necessary epigenetic modifications for CD4+CD8αα+ IEL differentiation. Secondly, it is possible that CCR9 overexpression in the absence of signaling partners, which are normally present in epithelial T cells, is not sufficient to interfere with this program. Because TCR engagement is essential for the induction of the CD4+CD8αα+ T cells[23], a third possibility is that alterations to the TCR repertoire could lead to an

enrichment of CD4+ T cell pools that are likely to differentiate into CD4+CD8αα+ IELs. Given that the homing capacity to the thymus is reduced by CCR9 deficiency[33], it is possible that CCR9-deficient thymic T cells undergo a distinct primal differentiation process in the thymus to compensate for the lower number of pre-thymic T cell progenitors. However, our TCR analysis suggests that the acquisition of CD4+CD8αα+ IEL characteristics through CCR9 deficiency is unlikely due to a biased TCR repertoire, and further analysis is needed to uncover the mechanism. Moreover, as CD4+ IELs developed differently between *Ccr9+/+* and *Ccr9−/−* mice based on our bioinformatics and trajectory analyses, further studies are needed to provide direct evidence on how CD4+ IELs in *Ccr9+/+* and *Ccr9−/−* mice were differentiated.

Recently some CCRs are involved in cell signaling. For instance, C-X-C motif chemokine ligand 12, a ligand for the C-X-C motif chemokine receptor 4, simultaneously activates calcium release, resulting in activation of the phosphatidylinositol 3-kinase (PI3K)/AKT and RAS/mitogen-activated protein kinase (MAPK) pathways[34]. The expression of CCR5 affects *Junb/Jund, Tbx21, Il2rb*, and *Fos* expression related to cell proliferation and activation of lung natural killer cells[35]. CCR9 receptor-mediated signaling activates PI3K/AKT in solid cancer; however, the effects of CCR9 receptor signals in CD4+CD8αα+ IELs are unknown, apart from the chemo-attractive effects towards CCL25[36–38]. As CD4+CD8αα+ IELs were induced in a CCL25-independent manner, CCL25-CCR9 signaling was not involved in the induction of the CD4+CD8αα+ T cells. Moreover, a lack of *Ccr9* in T cells increases the proportion of CD4+CD8αα+ T cells in IE compartment and LP with the expression of *Runx3, Cbfb*, and *CD8α*. CD4+CD8αα+ T cell development is not location-dependent, and gradually settles in the IE compartment under *Ccr9* deficiency. Taken together, loss of CCR9 expression may modulate intracellular signaling, altering gene expression profiles in the gut. *Cbfb2, Runx3*, and *Tbx21*, which promoted the differentiation to CD4+CD8αα+ IELs, were preferentially upregulated in *Ccr9−/−* mice rather than *Ccr9+/+* mice. Notably, these genes were upregulated after migrating to the gut, but the precise mechanism underlying differential CD4+CD8αα+ IEL development between *Ccr9+/+* and *Ccr9−/−* mice remains to be determined.

The induction of RUNX3 with downregulation of ThPOK is essential for the differentiation of CD4+CD8αα+ IELs[3]. All RUNX proteins (RUNX1−3) must associate with CBFβ to exert their transcriptional regulatory functions[39]. Unlike that of *Runx3*, the expression of *Cbfb* has not been thoroughly examined during the differentiation of CD4+CD8αα+ IELs. Here, we found that the CD4+CD8αα− IEL precursor population with low CCR9 expression tended to express higher levels of the *Cbfb2* variant, and *Cbfb2* transcription was increased during the differentiation of CD4+CD8αα+ IELs. In addition, genetic mutations

that abrogated *Cbfb2* splicing resulted in impaired CD4+CD8αα+ IEL differentiation. These observations revealed that the induction of *Cbfb2* splicing is an important regulatory process for CD4+CD8αα+ IEL differentiation. The loss of *Cbfb2* splicing is reported to impair the differentiation of lymphoid tissue inducers, Langerhans, and dendritic epidermal T cells, indicating that *Cbfb2* splicing is crucial for the differentiation of several types of cells[40,41]. Given the distinct C-terminal sequences between Cbfβ1 and Cbfβ2, it is possible that Cbfβ2 plays a unique role in regulating the expression of target genes via the RUNX/CBFβ complex. Alternatively, *Cbfb2* splicing may primarily function to increase the total amount of CBFβ protein. The frequency of CD4+CD8αα+ IELs was low but still observable in *Cbfb2m/2m* mice. Thus, it is conceivable that Cbfβ1 and Cbfβ2 may act in a complementary manner to support CD4+CD8αα+ IEL differentiation. Further investigation is needed to determine whether upregulation or deletion of *Cbfb1* affects CD4+CD8αα+ IEL differentiation.

In our analyses, we discovered a negative correlation between *Ccr9* and *Cbfb2* expression in IELs. However, we observed no difference in splenic *Cbfb2* expression between *Ccr9+/+* and *Ccr9-/-* mice. This implies that CCR9 signaling alone does not modulate *Cbfb2* splicing, and that IE-specific factors may be involved. The mechanism behind *Cbfb* splicing regulation is not well understood, making it unclear why *Cbfb2* splicing was increased in the IELs of *Ccr9-/-* mice while *Cbfb1* splicing remained unchanged during the differentiation of CD4+CD8αα+ IELs.

Epigenetic changes have been shown to occur during the differentiation of CD4+CD8αα+ IELs, and the intestinal epithelial environment is important for the acquisition of IEL-related epigenetic profiles[5]. RA is an important factor for the differentiation of CD4+CD8αα+ IELs and induces epigenetic changes in Treg cells, and in Th1 and Th17 cells[3,9,42,43]. However, the genomic binding pattern of CBFβ2 detected by ChIP-seq was different between spleen cells and IELs, indicating that factors other than RA in the intestinal environment are required to induce epigenetic changes in CD4+ T cells and affect the gene binding pattern of CBFβ2. Although the exact mechanism of the interaction between Treg cells, Th1 cells, and *Cbfb2* gene is still unclear, *Cbfb2* overexpression alters the proportion of CD4+ T cells.

Although the function of human CD4+CD8αα+ T cells is not well elucidated, their reduced abundance in inflammatory bowel diseases has been reported[44]. Our findings regarding CD4+CD8αα+ T cells with low CCR9 expression provide insights into a potential target to maintain gut homeostasis.

## Methods

### Animals
C57BL/6J mice were purchased from CLEA Japan (Tokyo, Japan), and C57BL/6J-*Ccr9-/-* mice were purchased from Sankyo Labo Service Corporation (Tokyo, Japan). VA deficiency was established by administering a VA-deficient diet obtained from CLEA Japan. C57BL/6J-*Ly5.1*, C57BL/6J-*Rag2-/-*, and germ-free mice were purchased from Sankyo Labo Service Corporation. C57BL/6J-*Thpok*GFP:*Runx3*tdTomato reporter, C57BL/6J-*Cbfβ2m/2m*, *Rosa26*lsl-cbfb2-GFP, and C57BL/6J-*Cd4-Cre* mice were produced as previously described[14,41,45,46]. The C57BL/6J-*Rosa26*lsl-Ccr9-GFP mouse strain was generated by homologous recombination in embryonic stem cells (ESCs) transfected with a vector (Eurofins Genomics, Luxembourg, Luxembourg) harboring complementary DNA (cDNA) encoding a *Ccr9* insertion at the *AscI* site. The detailed procedure for generating *Cbfb1*tdTomato and *Cbfb2*Venus reporter alleles was conducted as follows: cDNA encoding tdTomato or Venus was targeted to exon 6 in the *Cbfb* locus of ESCs by homologous recombination. C57BL/6J-*Ccl25+/+* and C57BL/6J-*Ccl25-/-* mice were provided by Dr. Sayama (Shizuoka University, Japan). Mice were analyzed at 7–12 weeks old age and male. 17 weeks old male C57BL/6J mice were used for cell sorting to perform ChIP-seq. Mice, except for the germ-free mice, were maintained under specific pathogen-free conditions with a 12-h light/dark cycle, at a temperature or 22–25 °C and a relative humidity of 45–55% in the Animal Care Facility of Keio University School of Medicine. Germ-free mice were bred and maintained in vinyl isolators. Littermate animals were used as controls. All experiments were approved by the regional animal study committees and were performed according to Keio University institutional guidelines (D2006-008).

### Preparation of IELs and LP mononuclear cells
Mice were euthanized by cervical dislocation and the small intestine was excised and opened longitudinally; Peyer's patches were removed. The small intestine was then washed with calcium- and magnesium-free Hank's balanced salt solution (HBSS) (17460-15; Nacalai Tesque, Kyoto, Japan) to remove fecal content. After washing, the small intestine was cut into small pieces and incubated with HBSS containing 1 mM dithiothreitol (15508-013; Invitrogen, Thermo Fisher Scientific, Waltham, MA, USA) and 5 mM EDTA (15575-038; Invitrogen) for 30 min at 37 °C to remove the epithelial layer. The mucosal pieces were then washed with HBSS and dissolved in solution by incubation with HBSS containing 1.5% fetal bovine serum (FBS; 10270-106; Thermo Fisher Scientific), 1.0 mg/mL collagenase (032-22364; Wako Pure Chemical, Tokyo, Japan), and 0.1 mg/mL DNase (DN25-1G; Sigma-Aldrich) for 30 min at 37 °C. The dissolved solution was centrifuged and the resulting pellet was resuspended in 40% Percoll (17-0891-01; GE Healthcare, Chicago, IL, USA) and overlaid on 75% Percoll (GE Healthcare, Chicago, IL, USA). Percoll gradient separation was performed by centrifugation at $840 \times g$ and 20 °C for 20 min. LP mononuclear cells were collected at the interphase between 40 and 75% Percoll layers. For IELs, the supernatant containing the epithelial layers was collected and centrifuged. The resulting pellet was resuspended, separated, and collected following the procedures used for mucosal tissue.

### Preparation of MLN and spleen cell suspensions
MLN and spleen tissues were harvested from mice after euthanasia and homogenized manually in HBSS. The lysates were filtered through a cell strainer. MLN cells were centrifuged and collected. Splenic cells were hemolyzed with 0.84% (v/w) ammonium chloride (02424-55; Nacalai Tesque), washed with HBSS, and collected for analysis.

### Flow cytometry
After blocking with anti-FcR (553141, BD Biosciences, NJ, USA) for 5 min, the cells were incubated with specific fluorescence-labeled Ab and/or 7-AAD (51-68981E, BD Biosciences)/Fixable viability dye (65-0865-14, Thermo Fisher Scientific, Tokyo, Japan). The surface antigens of isolated single-cell suspensions were stained with the following antibodies: CD4 (BioLegend, APC/BV421, RM4-5), CD8α (BD Biosciences, V500/PE-Cy7, 53-6.7), CD8β (Thermo Fisher Scientific, FITC/APC, eBioH35-17.2,), CD45 (BioLegend, BV510, 30-F11), CD45.1 (BD Biosciences, FITC, A20), CD45.2 (BioLegend, BV510/PE-Cy7, 104), TCRβ (BioLegend, APC-Cy7/PE-Cy7, H57-597), TCRγδ (BioLegend, PerCP-Cy5.5, GL3), α4β7 (BD Biosciences, PE, DATK32), CCR9 (BD Biosciences), BV421, CW-1.2). For intracellular Foxp3 (Thermo Fisher Scientific, PE/PerCP-Cy5.5, FJK-16s), and Ki67 (BioLegend, PE, 16A8) staining, cells were permeabilized using a fixation/permeabilization solution (00-5523-00; Thermo Fisher Scientific) before intracellular staining with antibodies. For IL-17A (Thermo Fisher Scientific, PE, eBio17B7,) and IFN-γ (BD Biosciences, PE-Cy7, XMG1.2) staining, cells were incubated in RPMI-1640 medium containing 10% FBS and 1% penicillin/streptomycin (09367-34; Nacalai Tesque) with 50 ng/mL phorbol-12-myristate-13-acetate (P1585-1mg; Sigma-Aldrich), 500 ng/mL ionomycin (I0634-1mg; Sigma-Aldrich), and GoldiStop Protein Transport Inhibitor (554724, BD Bioscience). Cells were stained with antibodies against the indicated cell surface markers, and dead cells were stained with eFluor780 fixable viability dye. Cells were permeabilized using a fixation/permeabilization solution before IL-17A and

IFN-γ staining. Splenic cell suspensions were stained with the following antibodies: CD45 (BioLegend, BV510, 30-F11), TCRβ (BioLegend, APC-Cy7/PE-Cy7, H57-597), CD4 (BioLegend, APC/BV421, RM4-5), CD8α (BD Biosciences, V500/PE-Cy7, 53-6.7), CD44 (BioLegend, APC, IM7), and CD62L (BioLegend, PerCP-Cy5.5, MEL-14). Flow cytometry antibodies were purchased from BD Biosciences, BioLegend, and Thermo Fisher Scientific. For flow cytometry analysis, the fluorescence-activated cell sorting (FACS) Canto II system (BD Biosciences, Franklin Lakes, NJ, USA) was used. Data were analyzed using FlowJo Vx software (Tree Star Inc., OR, USA). A FACS Aria III system (BD Biosciences) was used for cell sorting. The gating strategy for FACS analysis is presented in Supplementary Fig. 10. Detailed information for the antibodies used in this study is summarized in Supplementary Data 3.

### In vitro CD4⁺CD8αα⁺ T cell differentiation

Naive CD4⁺ T cells were isolated using a MACS CD4⁺CD62L⁺ isolation kit (130-106-643; Miltenyi Biotec, Bergisch Gladbach, Germany). A total of $1 \times 10^5$ naive CD4⁺ T cells were cultured in T cell culture medium containing 10% FBS, 1% penicillin/streptomycin, 1% pyruvate (11360070; Thermo Fisher Scientific), 1% minimum essential medium/non-essential amino acids (11140050; Thermo Fisher Scientific), 2.5% HEPES (15630080; Thermo Fisher Scientific), and 55 mM β-mercaptoethanol (21985023; Thermo Fisher Scientific). Naive CD4⁺ T cells were cultured with plate-bound anti-CD3e (1 μg/mL, 100359; BioLegend) and anti-CD28 (1 μg/mL, 102116; BioLegend), RA (10 nM, 0064-1 G; Tokyo Chemical Industry, Tokyo, Japan), TGF-β (2 ng/mL, 7666-MB-005; R&D Systems, Minneapolis, MN, USA), IFN-γ (20 ng/mL, 315-05; PeproTech, Cranbury, NJ), and TCDD (31 nM, 48599; Sigma-Aldrich) in presence or absence of CCL25 (100 ng/mL, 481-TK-025; R&D systems).

### Nucleofection

Naive CD4⁺ T cells were obtained from $Ccr9^{+/+}$ and $Ccr9^{-/-}$ mice as described in "In vitro CD4⁺CD8αα⁺ T cell differentiation." A total of $1 \times 10^6$ naive CD4⁺ T cells were resuspended in 20 μL primary cell nucleofector solution (P4 Primary Cell 4D-Nucleofector X Kit S [32RCT, V4XP-4032], Lonza) with 5 nM siRNA targeting negative control, $Ccr9$, $Runx3$, $Cbfb$, and $Tbx21$ (scramble, siCcr9, siRunx3, siCbfb, and siTbet, respectively). Cells were transferred to nucleofection cuvette strips and were electroporated using a 4D nucleofector (4D-Nucleofector Core Unit: AAF-1002B; 4D-Nucleofector X Unit: AAF-1002X; Lonza, Gampel). CM137 pulses were performed, and cells were incubated for 10 min at 25 °C after nucleofection. Transfected cells were transferred to 96-well plates containing 37 °C T cell medium with β-mercaptoethanol and then incubated at 37 °C and 5% $CO_2$ for 3 h. Cells were washed once with T cell culture medium, and nucleofected cells were cultured as described in "In vitro CD4⁺CD8αα⁺ T cell differentiation."

The following siRNA sequences were used: siCcr9 sense, 5′-UGAUAAACAUUCUAAUGCAtt-3′, and antisense, 5′-UGCAUUAGAAUGUUUAUCAag-3′; siRunx3 sense, 5′-CACCAACCUUCAUACGAGAtt-3′, and antisense, 5′-UCUCGUAUGAAGGUUGGUGta-3′; siCbfb sense, 5′-GAAGAACUCGAGAAUUUGAtt-3′, and antisense, 5′-UCAAAUUCUCGAGUUCUUCtt-3′; and siTbet sense, 5′-GAUCAUCACUAAGCAAGGAtt-3′, and antisense, 5′-UCCUUGCUUAGUGAUGAUCat-3′. For the negative control, $Silence$ Select Negative Control No. 1 siRNA (4390843, Thermo Fisher Scientific) was used.

### Quantitative real-time PCR

RNA was isolated using TRIzol (15596018; Thermo Fisher Scientific) according to the manufacturer's instructions. cDNA was synthesized from the extracted RNA using an iScript cDNA Synthesis Kit (1708891; Bio-Rad, Hercules, CA, USA) and then amplified by quantitative real-time PCR using primer sets and the KAPA SYBR Green FAST qPCR Master Mix

Kit (KK4602; Roche, Basel, Switzerland). The housekeeping gene $Rpl32$ was used to normalize samples. The following primer sequences were used: $Rpl32$ forward, 5′-ACAATGTCAAGGAGCTGGAG-3′, and reverse, 5′-TTGGGATTGGTGACTCTGATG-3′; $Cbfb1$ forward, 5′-CGTAATGGAGTGTGTGTTAT-3′, and reverse, 5′-TTGCTGTCTTCTTGCCTCCA-3′; $Cbfb2$ forward, 5′-CGTAATGGAGTGTGTGTTAT-3′, and reverse, 5′-TTGCTGTCTTCTTGCCAGTT-3′; $Runx3$ forward, 5′-GGTCACCACCGTTCCATC-3′, and reverse, 5′-ACTTCCTCTGCTCCGTGCT-3′; $Tbx21$ forward, 5′-ATCCTGTAATGGCTTGTGGG-3′, and reverse, 5′-TCAACCAGCACCAGACAGAG-3′; $Zbtb7b$ forward, 5′-ATGGGATTCCAATCAGGTCA-3′, and reverse, 5′-TTCTTCCTACACCCTGTGCC-3′; and $Ccr9$ forward, 5′-CAATCTGGGATGAGCCTAAACAAC-3′, and reverse, 5′-ACCAAAAACCAACTGCTGCG-3′; $Ccl25$ forward, 5′-TTACCAGCACAGGATCAAATGG −3′, and reverse, 5′-CGGAAGTAGAATCTCACAGCAC-3′.

### Fecal sample collection and bacterial DNA extraction

Fecal samples were collected from $Ccr9^{+/+}$ and $Ccr9^{-/-}$ mice and immediately frozen at −80 °C. The bacterial pellet was suspended and incubated with lysozyme (Sigma-Aldrich) at 37 °C for 1 h in TE10 (10 mM Tris-HCl, 10 mM EDTA, pH 8.0). Achromopeptidase (Wako Pure Chemical) was added, and samples were incubated at 37 °C for 30 min. The suspension was treated with 20% sodium dodecyl sulfate and proteinase K (Merck, Rahway, NJ, USA), incubated at 55 °C for 1 h, and treated with phenol/chloroform/isoamyl alcohol (Invitrogen). The DNA pellet was rinsed with 75% ethanol and dried. DNA samples were purified by RNase A (Wako) treatment and precipitated with 20% polyethylene glycol solution (20% PEG-2.5 M NaCl). DNA was then pelleted by centrifugation, rinsed with 75% ethanol, and dissolved in TE (10 mM Tris-HCl, 1 mM EDTA, pH 8.0) buffer.

### Metagenomic analysis of 16S rRNA

The hypervariable V3–V4 region of the $16S$ gene was amplified using Ex Taq Hot Start (RR006A; Takara Bio, Kusatsu, Japan) and purified using AMPure XP (A63881; Beckman Coulter). Mixed samples were prepared by pooling approximately equal amounts of each amplified DNA, and were sequenced using the Miseq Reagent Kit V3 (600 cycles) and the Miseq platform (both Illumina, San Diego, CA, USA), according to the manufacturer's instructions. Sequences were analyzed using the QIIME software package version 1.9.1[47,48]. Paired-end sequences were joined using the fastq-join tool in the ea-utils software package (ver 1.1.2)[49]. High-quality sequences for each sample (15,000) were randomly selected after quality filtering. After trimming off both primer sequences using cutadapt[50] and using de novo chimera detection using USEARCH[51], the sequences were assigned to operational taxonomic units (OTUs) using the UCLUST algorithm[52] with a sequence identity threshold of 96%. Taxonomic assignment of each OTU was determined via similarity searching against publicly available $16S$ data (RDP version 10.27 and CORE update September 2, 2012) and using the NCBI genome database with the GLSEARCH program.

### Quantitative PCR amplification of 16S rRNA gene

Quantitative PCR was performed using the CFX Opus 96 Real-Time PCR Instrument (Bio-Rad) with the KAPA SYBR Fast qPCR Master Mix Kit (KK4602, Roche). The primer pair "all bacteria," 5′-CGGTGAATACGTTCCCGG-3′ and 5′-TACGGCTACCTTGTTACGACTT-3′, was used.

### Antibiotic treatment

Ampicillin (6.7 mg/mL; A9518-25G, Sigma-Aldrich), neomycin (6.7 mg/mL; N1876-25G, Sigma-Aldrich), metronidazole (6.7 mg/mL; M3761-25G, Sigma-Aldrich), and vancomycin (3.35 mg/mL; 226-01306, Wako) were dissolved in sterile distilled water and administered to mice three times a week (500 μL per mouse) by oral gavage. The same volume of sterile distilled water was administered to the control group on the same schedule.

## Fecal microbiome transplantation

Fecal samples were collected from *Ccr9*[+/+] and *Ccr9*[−/−] mice derived from the same facility and stored at −80 °C until use. Feces was homogenized in phosphate-buffered saline (PBS) and filtered through a mesh strainer. The filtered content was orally administered to germ-free mice in the vinyl isolator. Three weeks after the fecal microbiome transplantation, mice were analyzed for FACS analysis.

## Single cell–based transcriptome and TCR repertoire analysis

CD4[+] T cells from the spleen, mesenteric lymph nodes (MLNs), LPLs, and IELs were sorted from *Ccr9*[+/+] and *Ccr9*[−/−] mice. Single-cell suspensions (*Ccr9*[+/+]; 8796 IELs, 3868 LPLs, 1106 MLNs, and 953 splenocytes, *Ccr9*[−/−]; 9353 IELs, 3860 LPLs, 976 MLNs, and 615 splenocytes) were loaded onto chromium microfluidic chips to generate single-cell gel-bead-in-emulsion using the chromium controller (10x Genomics, Pleasanton, CA, USA) according to the manufacturer's instructions. RNA from the barcoded cells for each sample was subsequently reverse-transcribed inside the gel-bead-in-emulsion using a C1000 Touch Thermal Cycler (Bio-Rad), and all subsequent steps to generate single-cell libraries were performed according to the manufacturer's protocol. Libraries were sequenced on an DNB-Seq G400 sequencer as paired-end mode (read1: 28 bp; read2: 100 bp). The raw reads were processed by Cell Ranger 7.0.1 (10x Genomics). The processed data were aggregated and analyzed by Seurat v 4.0.6 and scanpy (ver 1.9.1). The PCR amplified VDJ reads were mapped against the refdata-cell-ranger-vdj-GRCm38-alts-ensembl-5.0.0 mouse reference genome in Single Cell V(D)J R2-only chemistry mode. Hashtag oligo demultiplexing was performed using the Seurat HTODemux function. Only cells assigned a single hashtag and a beta-chain clonotype were retained for downstream analysis. CDR3 similarity (TCR sharing/clonal overlap) was calculated using the Morisita-horn index with the divo package (version 1.0.1), visualized by seaborn heatmap (ver v.0.11.2). Clonality, as indicated by Shannon's evenness index, was calculated using the skbio.diversity.alpha package (version.0.4.2).

## Normalization, sample integration, dimensional reduction

Gene expression–based clustering was performed using the scanpy package (ver 1.9.1)[53]. Cells with mitochondrial contents greater than 2%, with ribosomal contents greater than 30%, and with fewer than 700 genes and 1000 counts detected (dying cells, empty droplets, doublets, respectively) were considered outliers and were filtered out. Cells were log normalized, and BBKNN[54] was used to correct batch effects across the different samples. Integrated counts were scaled and centered for principal component (PC) analysis and dimensional reduction. For each scaled dataset, 50 PCs were calculated using the top 2000 most variable genes regressed out from the mitochondrial and ribosomal genes.

## Clustering and differential gene expression analysis

From the corrected integrated reduced PC analysis data, Leiden clustering analysis was performed for the first 10 PCs, with a nearest neighbor value of 10 and a minimum distance of 0.01. Clustering was performed with a resolution of 1, which resulted in 16 clusters that were combined and annotated into specific cell types based on known marker genes. The top marker genes for each defined celltype were estimated using a *t* test overestimated variable approach. Pairwise differential analysis between two clusters were generated using a Welch's *t* test approach from the diffxpy scanpy package and only those genes with significant log fold changes and adjusted *P* values (fold change >0.3, *P* < 0.05) were chosen. Gene expression was visualized using seaborn violinplot (ver 0.11.2).

## Single cell pseudotime analysis

To investigate transitions between T cell subsets, pseudotime trajectory analysis was performed (Monocle 3 ver 3.0) by splitting the

dataset into *Ccr9*[+/+] and *Ccr9*[+/+] and setting the naive CD4[+] T cell population as the root population for both mouse group.

## Bone marrow transplantation

Bone marrow cells were harvested from C57BL/6 *Cd45.1*[+]*Ccr9*[+/+] and *Cd45.2*[+] *Ccr9*[+/+] mice by gently flushing collected femurs with RPMI 1640 medium. A total of $6 \times 10^6$ cells (1:1 ratio of C57BL/6 *Cd45.1*[+]*Ccr9*[+/+] to *Cd45.2*[+]*Ccr9*[−/−] cells) were intravenously injected into C57BL/6 (CD45.2 background) recipient mice immediately after irradiation (11 Gy). Cells in recipient mice were analyzed 4 weeks after transplantation.

## T cell transfer model

CD4[+] T cells were obtained using L3T4 microbeads (130-117-043, Miltenyi Biotec). A total of $3 \times 10^5$ cells (1:1, $1.5 \times 10^5$ each from C57BL/6 *Cd45.1*[+]*Ccr9*[+/+] and *Cd45.2*[+]*Ccr9*[−/−] mice) were intraperitoneally injected into *Rag2*[−/−] mice. IELs in mice were analyzed 6 weeks after the transfer.

## Administration of EdU and analysis of EdU expression'

*Ccr9*[+/+] and *Ccr9*[−/−] mice were intraperitoneally administered 400 μg EdU (C10635, Thermo Fisher Scientific) daily for 7 days, and small intestine IELs were collected as described in the "Preparation of IELs and LP mononuclear cells" section. After surface and intracellular staining, the click iTR EdU reaction was performed to detect EdU expression according to the manufacturer's instructions.

## ChIP-seq

CD4[+]CD8αα[+] IELs, CD4[+]CD8αα[−] IELs, and splenic naive CD4[+] T cells from C57BL/6J mice were washed once with PBS containing 1% FBS and cross-linked by incubation in a 1% formaldehyde solution for 10 min with gentle rotation at 25 °C. The reaction was stopped with glycine solution (final concentration of 0.15 M), and the cells were lysed in lysis buffer 1 (50 mM HEPES, pH 7.5, 140 mM NaCl, 1 mM EDTA, 10% glycerol, 0.5% NP40, and 0.25% Triton X-100) supplemented with complete protease inhibitor cocktail tablets (Roche). The nuclei were then pelleted and washed with lysis buffer 2 (10 mM Tris-HCl, pH 8.0, 200 mM NaCl, 1 mM EDTA, and 0.5 mM EGTA) supplemented with complete protease inhibitor (Roche). Next, the nuclei were resuspended in lysis buffer 3 (10 mM Tris-HCl, pH 8.0, 100 mM NaCl, 1 mM EDTA, 0.5 mM EGTA, 0.1% sodium deoxycholate, and 0.5% *N*-laurylsarcosine sodium salt) and sonicated using an XL2000 ultrasonic cell disruptor (Qsonica, CT, USA) at output level 6 for 15 s. After the addition of Triton X-100, sonicated chromatin was incubated overnight at 4 °C in the presence of anti-Cbfβ2 antibody (made in the Taniuchi lab[55]) pre-conjugated with Dynabeads M-280 sheep anti-rabbit IgG (Thermo Fisher Scientific). After washing beads with ChIP-RIPA buffer (50 mM HEPES, pH 7.6, 500 mM LiCl, 1 mM EDTA, 1% NP-40, and 0.7% sodium deoxycholate) and TE buffer supplemented with 50 mM NaCl, immunoprecipitates were eluted from beads into elution buffer (50 mM Tris-HCl, pH 8.0, 10 mM EDTA, and 1% SDS) by incubation for 15 min at 65 °C. Eluted immunoprecipitates were then incubated at 65 °C overnight for reverse cross-linking. Input and ChIP DNA were treated with Rnase A (Thermo Fisher Scientific) and Proteinase K (Thermo Fisher Scientific). DNA was extracted with phenol/chloroform, and purified DNA was subjected to re-sonication in a Covaris S220 sonicator to produce DNA fragments with a mean size of 200 bp. These fragments were used for library construction with the NEBNext ChIP-seq Library Prep Master Mix set for Illumina kit (New England Biolabs, Ipswich, MA, USA). Sequencing was performed at the RIKEN IMS sequencing facility (Yokohama, Japan) with an Illumina HiSeq 1500 instrument. Sequence reads were aligned on the mm9 mouse genome using bowtie2 (v.2.1.0, http://bowtie-bio.sourceforge.net /index.shtml) with default parameters. Peaks were called using Homer (version 4.10) with the default parameters.

## Statistical analysis

Statistical analyses were performed using GraphPad Prism version 6 (GraphPad software Inc. CA, USA). Statistical tests and corresponding *n* values are reported in each figure legend. Data from FACS and quantitative real-time PCR were expressed as means ± standard deviation (SD) or standard error of the mean (SEM). Differences between two groups were evaluated using two-sided unpaired Student's *t* tests. Comparisons of more than two groups was performed with one-way ANOVA followed by Tukey–Kramer's multiple comparison test. *P* values of <0.05 were considered significant. Each experiment was replicated more than two times with reproducible results.

## Reporting summary

Further information on research design is available in the Nature Portfolio Reporting Summary linked to this article.

## Data availability

The data for ChIP-seq and scRNA-seq have been deposited in DNA Data Bank of Japan. Accession numbers are DRA014018 for ChIP-seq, and DRA015828 and DRA014019 for scRNA-seq. ChIP-seq based on mm9 mouse genome reference (https://hgdownload.soe.ucsc.edu/goldenPath/mm9/chromosomes/). scRNA-seq based on mm10 mouse genome reference (https://hgdownload.soe.ucsc.edu/goldenPath/mm10/chromosomes/). All other data are available in the article and its Supplementary files or from the corresponding author upon request. Source data are provided with this paper.

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

## Acknowledgements

We thank A. Yoshimura (Department of Immunology and Microbiology, Keio University) for providing valuable comments. We also thank A. Tojo, A. Chida, S. Umeda, E. Nomura, T. Suzuki, H. Nakatsukasa, S. Yamamoto (Keio University School of Medicine), and B. Reis and TBR Castro (The Rockefeller University) for their assistance with the experiments. We are grateful to K. Yamashita (KOTAI Biotechnologies, Inc. Osaka, Japan) for supporting scRNA-seq data analysis. Finally, we thank Editage for editing a draft of this manuscript. This work was supported by Grants-in-Aid from the Japanese Society for the Promotion of Science (JSPS) (19K22624, 17H05082, 19K22624, 20H03665, 21K18272, and 23H02899 to T.S.; 21K07084 to K.O.; 19K08402 to N.H.; 21H02905 to H.O.; 20H00536 and 23H00425 to T.Kanai), JST Forrest (21457195 to T.S.), the Japan Agency for Medical Research and Development (19ek0109214 to T.S.; CREST-16813798, and JP21gm1510002 to T.Kanai), the Mochida Memorial Foundation 2017, 2021 (to T.S.), the Takeda Science Foundation (to T.S.), the GSK Science Foundation (to T.S.), the Yakult Bioscience Research Foundation (to T.S.), the Keio University Fukuzawa Foundation (to T.S.), the Keio University Grant-in-Aid for Encouragement of Young Medical Scientists 2017, 2018 (to K.O.), and the Mitsubishi Foundation (to T.Kanai).

## Author contributions

Conceptualization: T.S.; methodology: K.O.; investigation: K.O., K.M., Y.H., S.K., Y.Y., S.T., Y.K., J.Z., K.S., and T. Koide; visualization: K.O. and K.M.; funding acquisition: K.O., T.S., N.H., and H.O. and T. Kanai; supervision: T.S. and T. Kanai; writing—original draft: K.O. and T.S.; writing—review and editing: T.T., Y.M., K. T., N.N., M.L., D.M., and I.T.

## Competing interests

The authors declare no competing interests.
K.M. is an employee of Miyarisan Pharm. Y.K. is an employee of Tanabe Mitsubishi Pharm. The other authors declare no competing interests.
