## [Peer Review File · Nature Communications]

Downregulation of chemokine receptor 9 facilitates CD4 +CD8 $\alpha\alpha$ + intraepithelial lymphocyte developmentREVIEWER COMMENTS

Reviewer #1 (Remarks to the Author):

The manuscript by Ono et al examines the importance of CCR9 in regulating the small intestinal CD4+ intraepithelial lymphocyte (IEL) compartment of mice. While there are relatively few CD4+ T cells within the IEL compartment and their role in vivo is the subject of continued investigation, upon entry into the epithelium these cells take on CD8+ T cell-like characteristics including upregulation of the TF Runx3, downregulation of ThPOK and expression of the CD8 $\alpha\alpha$ homodimer.

The authors show that absence of CCR9 alters the composition and transcriptional profile of CD4+ IEL, with enhanced populations of CD4+CD8 $\alpha\alpha$ + IEL. They further demonstrate a role for Cbf β 2, a Runx3 partner, in the generation of CD4+CD8 $\alpha\alpha$ + IEL. Interestingly, CCR9^{-/-} CD4+ T cells displayed an enhanced ability to convert to CD4+CD8 $\alpha\alpha$ + IEL-like cells and siRNA mediated knockdown of CCR9 enhanced the ability of CD4+ T cells to convert to CD4+CD8 $\alpha\alpha$ + IEL-like cells in vitro. Unfortunately, the mechanisms underlying this potentially important observation, that could explain the altered IEL composition of CCR9^{-/-} mice, was not sufficiently explored. While I found the manuscript to have merits, the results text was difficult to follow in several places with brief and often unclear descriptions. Further, I do not feel that the data is sufficiently robust to support several of the authors conclusions.

Major points.

Figure 1 and supplementary Figure 1.

The authors write in the title of Figure 1 'The proportion of CCR9 positive cells in CD4+CD8 $\alpha\alpha$ + IELs is less compared with other IEL population'. To me other IEL have a similar reduction in CCR9 compared to their LP counterparts. Can the authors rule out that the variations in CCR9 expression are not due to ligand (CCL25) dependent downregulation

of CCR9 as this might also explain why CCR9 levels are higher on the presumably more recently entered putative CD4⁺ IEL precursors of CD4⁺CD8 $\alpha\alpha$ ⁺ IEL. The authors also show data on CD4⁺FOXP3⁺ IEL. Is this really a true population of IEL as opposed to contaminants from the LP as they do not appear in the later scRNA-seq data sets. Can this population be identified in the epithelium of WT mice? To put these populations into better context for the reader this reviewer feels it is important to show the total number of all IEL subsets. While ThPOK^{hi} Runx3^{hi} CD4⁺ IEL express higher levels of CCR9 mRNA than ThPOK^{hi} Runx3^{low} cells (panel C) this does not appear to be reflected at the protein level where CCR9^{lo} and CCR9^{mid} cells contain similar proportions of these cells. In this regard the splitting of CCR9 expressing cells into low, mid and high cells appears very arbitrary and not convincing, with the low and high cells representing only a tiny proportion of cells (Figure S1C and D) with overall levels of CCR9 surface protein being highly similar between these three IEL populations. Further the staining in Figure S1C and D looks nothing like the CCR9 staining in Figure 1 A.

Figure 2 and supplementary Figure 2.

The increased proportions of CD8 $\alpha\alpha$ expressing CD4⁺ IEL in CCR9^{-/-} mice, in particular in young mice, is striking. Given the role of the microbiota in the generation of these cells are these comparisons performed using littermates or at least with cohoused animals?

The data in D and E suggests that this is due to an intrinsic enhanced ability of CCR9^{-/-} CD4⁺ T cells to convert to CD4⁺ CD8 $\alpha\alpha$ ⁺ T cells. Do these cells also upregulate Runx3 and downregulate ThPOK? Since there is no CCR9 ligand (CCL25) in these cultures what do the authors propose is the underlying (and potentially important) mechanism here, as it may explain the IEL phenotype? Given the importance of CCR9 in the thymus, do the authors believe it is an altered education of developing thymocytes in CCR9^{-/-} mice?

The results of the mixed BM chimera experiments in F and G are highly surprising given that multiple studies have demonstrated that CCR9^{-/-}BM is severely disadvantaged in generating peripheral CD4⁺ T cells in competitive settings with WT BM as the development of these cells is outcompeted within the thymus (see for example doi.org/10.1002/eji.200535203). Is

this selective for the CD4+ IEL population and if so what are the authors explanation for this discrepancy?

What is the authors explanation for the reduced numbers of FoxP3+CD4+ Tregs in the epithelium of CCR9-/- mice.

Figure 3 and supplementary Figure 3.

As far as I can tell the scRNA-seq analysis was performed on one WT and one CCR9-/- mouse. If this is the case this should be repeated as this is insufficient to draw any firm conclusions on differences between these strains. Were these littermates or at a minimum co-housed?

The authors identified three CD4+ IEL clusters however it is unclear why they do not detect FOXP3+ Treg IEL identified in Figure 1 and 2. To this reviewer the major potential finding from this data is that Cluster 3 is completely absent from the IEL compartment of CCR9-/- mice but for some reason this observation was not further explored (see below). What are these cells and can the authors confirm their absence in repeat scRNA-seq analysis or by flow cytometry?

Figure 4 and supplementary Figure 4.

In Figure 4A, the resolution is so poor that it was not possible to see where clusters 0,1 and 3 lie in the trajectory. The authors state 'The trajectory analysis indeed showed that Cluster 1 preceded Clusters 0 and 3 in CCR9+/+ mice and directly preceded Cluster 0 in CCR9-/- mice (Fig.4a), supporting that cells in cluster 1 were CD4+CD8 $\alpha\alpha$ + IEL precursors'. To this reviewer this argument is circular as the authors must have set cluster 1 as their pseudo-time starting point or? If cluster 3 is clonally unrelated to cluster 1 and 0 what is the authors rationale for including this population in the trajectory analysis? It has also been suggested that FOXP3+ Tregs are the immediate precursor of at least a proportion of CD4+CD8 $\alpha\alpha$ + IEL with regulatory function(DOI: 10.1126/science.aaf3892). Do such cells connect with CD4+CD8 $\alpha\alpha$ +

IEL in trajectory analysis?

For comparing gene between clusters and between WT and CCR9^{-/-} mice it is not appropriate to perform statistical analysis between one WT and one CCR9^{-/-} mouse (Figure 4C, Figure S4I). For the PCR analysis in Figure 4D, the enhanced expression of Runx3, Tbx21 and Cbfp2 in CCR9^{-/-} CD4⁺CD8 $\alpha\alpha$ ⁺ and CD4⁺CD8 $\alpha\alpha$ ⁻ IEL could simply be that these mice lack cluster 3 cells which express reduced levels of these markers? Indeed there appears to be no difference in Cbfb expression when comparing within the same cluster between CCR9^{-/-} and WT mice.

Figure 6.

While the results regarding Runx3, Cbfb and Tbet knockdown are somewhat expected, the result that knockdown of CCR9 enhances the ability of CD4⁺ T cells to convert into CD4⁺CD8 $\alpha\alpha$ ⁻ IEL-like cells is very interesting but not explored. What are the underlying mechanisms to this observation, in particular as the ligand CCL25 appears to be absent. What happens when CCL25 is added to the in vitro cocktail? What impact does CCL25 have on the transcriptional profile of these cells and are there differences in the transcriptional profile of CCR9^{-/-} and WT CD4⁺ T cells at the start and after 3 days of culture. This system would appear ideal for identifying the potential mechanisms driving CD4⁺ IEL alterations in CCR9^{-/-} mice.

Minor points

The authors write 'Collectively, these results showed that upregulation of Cbfb β in CD4⁺ T cells is essential for CD4⁺CD8 $\alpha\alpha$ ⁺ IELs differentiation'. Please tone this down as CD4⁺CD8 $\alpha\alpha$ ⁺ IELs are clearly present in Cbfb β deficient mice.

Similarly, in the results heading the authors write 'Downregulation of CCR9...' and 'CCR9 downregulation...' when comparing CCR9^{-/-} to WT mice and these titles should be changed to more accurately reflect what they are looking at.

Reviewer #2 (Remarks to the Author):

Summary: The investigators found that CCR9 is downregulated during the differentiation of Tregs to DP IELs, indicating that signaling through this receptor may limit DP IEL differentiation. This hypothesis was confirmed in CCR9-deficient mice, which exhibit an increase in DP IELs and LPLs. While the microbiota was shown to not regulate CCR9 expression, RA is required for the increased DP IEL differentiation in CCR9 KO mice. Moreover, CCR9 downregulation was shown to affect DP IEL precursors. Single-cell RNA sequencing revealed that Cbfb expression was increased in CCR KO mice, and further, Cbfb2 binds to Runx3 and Tbet promoter sites to regulate the induction of DP IELs. Lastly, in vitro experiments demonstrated that knockdown of CCR9, Cbfb2, Tbet and Runx3 are each required to facilitate DP IEL development.

Overall comments: This well-designed study makes use of a number of different mouse models to highlight a novel observation that downregulation of CCR9 is critical for the differentiation of DP IELs. The identification of DP IEL precursors and demonstration that Cbfb2 binds to Tbet and Runx3 promoters provides a convincing molecular mechanism. One key point that should be address relates to a more thorough explanation as to why DP lymphocytes are observed in the LP of CCR9 KO mice but not WT. Resolution of this question would allow the authors to strengthen their conclusion that DP lymphocyte development starts when the cells enter the epithelium. Additional clarifications and discussion relating to the potential mechanism by which CCR9 signaling would prevent DP IEL differentiation would provide a more complete story.

Major Concerns:

1. It is unclear why the gating of CCR9 differs from IELs and LPLs in Fig 1a and between spleen and MLN in Extended Figure 1b. Adding an FMO or isotype control would provide more confidence in the analysis and interpretation of the data presented.
2. CD4⁺ CD8^{aa}⁺ T cells develop in the epithelium and not LP in WT mice but are found in both locations in CCR9 KO mice. The authors conclude that the “developmental process does not start until the lymphocytes enter the epithelium”. First, it should be clarified that the prior statement applies to WT mice. Also, can the authors provide an explanation for

why there are DP T cells in the LP in the CCR9 KO? Is there a failure of CCR9 KO DP LPLs to migrate into the epithelium? Are there DEGs between the DP IELs and LPLs from CCR9 KO mice that could provide some insight?

3. Previous reports have characterized CCR9 downstream signaling in lymphocytes. While determining the exact mechanism restrains the DP IEL developmental program is beyond the scope of this study, it would be useful to discuss potential CCR9 downstream signals that may be involved.

Minor Concerns:

1. Page 6, the authors show that vitA, but not the microbiota is required for CCR9 expression; however, the conclusion states that “like microbiota, enhanced CD4⁺ CD8^{aa+} IEL differentiation in CCR9 KO mice is RA-dependent.” The way this is worded is confusing because the data show that the microbiota is not required for the phenotype observed in CCR9 KO mice.

2. The authors demonstrate that the frequency of IFN γ -expressing cells is decreased in the CD4-Cbfb2Tg mice and conclude that there are decreased Th1 LPLs, yet this point is not discussed further in the discussion.

Reviewer #3 (Remarks to the Author):

The authors investigated the effects of CCR9 expression and T cell differentiation. Overall, the manuscript was very difficult to read due to the wording of sentences and changing of subjects between connecting sentences. Additionally, it was not well defined why CCR9 is of importance.

For instance in the abstract, the second and third sentence seem to contradict one another. It is mentioned T cell differentiation can develop at the epithelium by the lamina propria and expression of CCR9 is low. The problem of interest is to define the mechanism preventing differentiation in LP but then later states that low CCR9 expression is found in

epithelial IELs and results in over differentiation in both epithelium and LP. How does CCR9 prevent differentiation in LP if it is shown to have over differentiation?

1. Third paragraph of the introduction: The first two sentences describe the expression of different subjects (gene or cells) making it difficult to understand. The first sentence describes CCR9 expression is down-regulated in CD4+ differentiation but the next sentence switches to IELs differentiation is increased in a different cell type, CD4+CD8+.

2. In the 'Loss of CCR9 enhances CD4+CD8+ IEL differentiation' section, paragraph 4: "Thus while downregulation of CCR9 strongly facilitates CD4+CD8+ cell differentiation, CCR9 overexpression is not sufficient to prevent this process.

The overexpression of CCR9 does not decrease CD4+CD8+ IELs differentiation. What is the importance in studying CCR9?

3. A model figure of the cell differentiation process and the experimental design would be extremely helpful.

4. In the single-cell RNA experiment, the pseudotime analysis is not described in the text. The statement " These data support that CD4+CD8+ T cells development starts in the intraepithelial compartment of Ccr9 -/- mice " is incorrect with only the cell type analysis. This is stated for Figure 3.

5. What are the cell number counts per tissue type? Were more cells sequenced for a particular type?

6. Figure 4A is not described in the main body of the text.

7. The scRNAseq data shows CD4+CD8+ IELs in Ccr9 -/- and Ccr9+/+ mice to be identical. This should be the expected result since the only change is to knock out Ccr9. The next statement, "Overall, the above data suggest that clonal expansion of CD4+CD8- IEL precursors in Ccr9-/- mice likely takes place after migration to the IE compartment", switches from CD4+CD8+ to CD4+CD8- cells. Also, how does this data suggest where clonal

expansion occurs? Is this part of the pseudotime analysis that is not mentioned?

8. Discussion. “ Our results indicate that CCR9 loss functionally facilitates CD4+ T cell precursors to acquire an IEL program..” How was this shown when overexpression also shows differentiation?

9. The figures are small and hard to read.

10. Data availability statement: Is there a dataset identifier from the DNA Data Bank of Japan?

REVIEWER COMMENTS

Reviewer #1 (Remarks to the Author):

Figure 1 and supplementary Figure 1.

The authors write in the title of Figure 1 ‘The proportion of CCR9 positive cells in CD4⁺CD8 $\alpha\alpha$ ⁺ IELs is less compared with other IEL population’. To me other IEL have a similar reduction in CCR9 compared to their LP counterparts. Can the authors rule out that the variations in CCR9 expression are not due to ligand (CCL25) dependent downregulation of CCR9 as this might also explain why CCR9 levels are higher on the presumably more recently entered putative CD4⁺ IEL precursors of CD4⁺CD8⁺ IEL.

The reviewer has made an important point regarding CCR9 as a gut homing receptor that is expressed at high levels in LP lymphocytes. Previous research has shown that CD4⁺CD8 $\alpha\alpha$ ⁺ IELs develop from CD4⁺CD8 $\alpha\alpha$ ⁻ IELs and ex-Foxp3 positive cells²⁻⁵. To further understand the role of CCR9 in IEL development, we evaluated the expression of CCR9 in CD4⁺CD8 $\alpha\alpha$ ⁺ and CD4⁺CD8 $\alpha\alpha$ ⁻ IELs and Tregs in the IE compartment. Additionally, we investigated whether the interaction between CCR9 and CCL25 affects the development of CD4⁺CD8 $\alpha\alpha$ ⁺ IELs. Our findings showed that the expression of *Ccl25* in the ileum was higher than that in the jejunum (R-Fig.1a). However, the proportion of CD4⁺CD8 $\alpha\alpha$ ⁺ IELs was comparable between the ileum and jejunum in 10-week-old *Ccr9*^{+/+} mice (R-Fig.1b).

We also observed that the expression of CCR9 was cell-dependent and not influenced by location, such as the jejunum and ileum. Therefore, these results suggest that the expression of CCR9 is dependent on cell type rather than the expression of CCL25 or location in the jejunum or ileum (R-Fig.1c,d : Fig. 1a,b).

We conducted an analysis of *Ccl25*^{+/+} and *Ccl25*^{-/-} mice to investigate whether CCL25 regulates CD4⁺CD8 $\alpha\alpha$ ⁺ IEL differentiation. Our findings indicate that the proportion of CD4⁺CD8 $\alpha\alpha$ ⁺ IELs was similar between *Ccl25*^{+/+} and *Ccl25*^{-/-} mice (R-Fig.1e). Based

on these results, we can conclude that CCL25, in contrast to its main receptor CCR9, does not play a role in the differentiation process of CD4⁺CD8 $\alpha\alpha$ ⁺ IELs.

R-Fig.1

(a) Relative expression of *Ccl25* in the jejunum and ileum. (b) Frequency of the CD8 α ⁺ population among TCR β ⁺CD4⁺CD8 β ⁻ IELs in the jejunum and ileum. (c) Histograms showing the CCR9 level among TCR β ⁺CD4⁺CD8 α ⁺CD8 β ⁻ Foxp3⁻ (CD4⁺CD8 $\alpha\alpha$ ⁺; red line), TCR β ⁺CD4⁺CD8 α ⁻CD8 β ⁻ Foxp3⁻ (CD4⁺CD8 $\alpha\alpha$ ⁻; green line), and TCR β ⁺CD4⁺CD8 α ⁻CD8 β ⁻ Foxp3⁺ (Tregs; blue line) IELs and LPLs in the jejunum and ileum Fluorescence minus one (FMO) control is shown as black line. (d) Mean fluorescence intensity (MFI) of CCR9 among CD4⁺CD8 $\alpha\alpha$ ⁺ and CD4⁺CD8 $\alpha\alpha$ ⁻ cells, Tregs, and FMO control IELs and LPLs. (e) Graphs show the frequency of CD8 α ⁺ or Foxp3⁺ cells among TCR β ⁺CD4⁺CD8 β ⁻ SI IELs of *Ccl25*^{+/+} and *Ccl25*^{-/-} mice. Data are expressed as mean \pm SD of individual mice (n=4-5, 2 pooled independent experiments). Statistical analysis was performed by Student's t-test (a, b, e), one-way analysis of variance

(ANOVA), and Tukey's post test (d). * $P < 0.05$, ** $P < 0.01$, *** $P < 0.001$, **** $P < 0.001$.

The authors also show data on CD4+FOXP3+ IEL. Is this really a true population of IEL as opposed to contaminants from the LP as they do not appear in the later scRNA-seq data sets. Can this population be identified in the epithelium of WT mice? To put these populations into better context for the reader this reviewer feels it is important to show the total number of all IEL subsets.

Using gene reporter techniques and live multi-photon imaging, we have previously shown that CD4⁺Foxp3⁺ T cells, despite representing a minor population in the epithelium, can be found in the gut IE compartment⁵. Additionally, scRNA-seq data indicates that cluster 11 corresponds to the Treg population, with cells present in the IE compartment and the LP (R-Fig. 6a-d: Fig. 3a-e). This suggests that Tregs among IELs are a true population. We have added the total number of IELs in all subsets. Our findings indicate that the total number of IELs, including CD4⁺ IELs, was similar between *Ccr9*^{+/+} and *Ccr9*^{-/-} mice at 7 and 10 weeks of age (R-Fig. 2b). As previously reported, the total number of TCRγδ⁺ cells in *Ccr9*^{-/-} mice was lower than that in *Ccr9*^{+/+} mice at both 7 and 10 weeks of age (R-Fig. 2a,b). However, the total number of CD4⁻CD8αβ⁺ T cells was comparable between *Ccr9*^{+/+} and *Ccr9*^{-/-} mice at 7 and 10 weeks of age (R-Fig. 2a,b). The increased number of CD4⁺CD8αα⁺ IELs in *Ccr9*^{-/-} mice at 7 and 10 weeks of age was not due to an increased total number of CD4⁺ T cells among IELs (Fig. 2b, c).

R-Fig. 2

(a) Frequency of CD8αβ⁺ and CD8αα⁺ populations among CD45⁺TCRβ⁺ cells and the TCRγδ⁺ population among CD45⁺ SI IELs. (b) Quantity of total IELs, CD4⁺, CD8αβ⁺, CD8αα⁺, and TCRγδ⁺ cells among SI IELs. Data are expressed as mean ± SD of individual mice (n=4–6, two pooled independent experiments). Statistical analysis was performed by one-way analysis of variance ANOVA, Tukey's post test (a, b). **P*<0.05, ***P*<0.01, ****P*<0.001, *****P*<0.0001.

While ThPOK^{hi} Runx3^{hi} CD4⁺ IEL express higher levels of CCR9 mRNA than ThPOK^{hi} Runx3^{low} cells (panel C) this does not appear to be reflected at the protein level where CCR9^{lo} and CCR9^{mid} cells contain similar proportions of these cells. In this regard the splitting of CCR9 expressing cells into low, mid and high cells appears very arbitrary and not convincing, with the low and high cells representing only a tiny proportion of cells (Figure S1C and D) with overall levels of CCR9 surface protein being highly similar between these three IEL populations. Further the staining in Figure S1C and D looks nothing like the CCR9 staining in Figure 1 A.

In our original submission, we utilized two different fluorophore panels to stain CCR9 in *ThPOK^{GFP}Runx3^{tdTomato}* mice and *Ccr9^{+/+}* mice (which lacked confounder reporters). However, we have now analyzed CCR9 expression in *ThPOK^{GFP}Runx3^{tdTomato}* mice using the same antibody panels used in *Ccr9^{+/+}* mice (R-Fig.3a-c). Our findings indicate that the population of CCR9^{hi} IELs is small, similar to the Treg population expressing CCR9 in IELs. As a result, we have removed the original Figures 1d and e and have replaced them with higher quality FACS plots (R-Fig. 3a : Fig. 1c). We emphasize that our objective is to demonstrate that CD4⁺CD8α⁺ IELs express lower levels of CCR9 than CD4⁺CD8α⁻ IELs, as this CCR9 expression is inversely correlated with the increased expression of Runx3, as we previously demonstrated⁴.

R-Fig.3

(a) Pseudocolor plots show three populations ($\text{ThPOK}^{\text{low}}\text{Runx3}^{\text{hi}}$, $\text{ThPOK}^{\text{hi}}\text{Runx3}^{\text{hi}}$, and $\text{ThPOK}^{\text{hi}}\text{Runx3}^{\text{low}}$) of CD4⁺ IELs and LPLs according to the expression of ThPOK and Runx3. (b) Histograms show the expression of CCR9 among $\text{ThPOK}^{\text{low}}\text{Runx3}^{\text{hi}}$ (red line), $\text{ThPOK}^{\text{hi}}\text{Runx3}^{\text{hi}}$ (green line), and $\text{ThPOK}^{\text{hi}}\text{Runx3}^{\text{low}}$ (blue line) populations of CD4⁺ IELs and LPLs. FMO control is shown as a black line. (c) Graphs show the MFI of CCR9 among $\text{ThPOK}^{\text{low}}\text{Runx3}^{\text{hi}}$, $\text{ThPOK}^{\text{hi}}\text{Runx3}^{\text{hi}}$, $\text{ThPOK}^{\text{hi}}\text{Runx3}^{\text{low}}$ and FMO control CD4⁺ IELs and LPLs. Data are expressed as mean \pm SD of individual mice (n=4, two pooled independent experiments). Statistical analysis was performed by one-way ANOVA, Tukey's post test (c). * $P < 0.05$, ** $P < 0.01$, *** $P < 0.001$, **** $P < 0.0001$.

Figure 2 and supplementary Figure 2.

The increased proportions of CD8 $\alpha\alpha$ expressing CD4⁺ IEL in CCR9^{-/-} mice, in particular in young mice, is striking. Given the role of the microbiota in the generation of these cells are these comparisons performed using littermates or at least with cohoused animals?

We assessed the percentage of CD4⁺CD8 $\alpha\alpha$ ⁺ IELs in cohoused littermates. The reviewer accurately noted that the microbiota makeup could impact the activation of CD4⁺CD8 $\alpha\alpha$ ⁺ T cells in *Ccr9*^{+/+} mice. To address this concern, we examined the microbiota composition in both strains and discovered no noticeable distinctions between *Ccr9*^{+/+} and *Ccr9*^{-/-} mice. These findings are now displayed in Supplementary Fig. 3h, i.

The data in D and E suggests that this is due to an intrinsic enhanced ability of CCR9^{-/-} CD4⁺ T cells to convert to CD4⁺ CD8 $\alpha\alpha$ ⁺ T cells. Do these cells also upregulate Runx3 and downregulate ThPOK? Since there is no CCR9 ligand (CCL25) in these cultures what do the authors propose is the underlying (and potentially important) mechanism here, as it may explain the IEL phenotype? Given the importance of CCR9 in the thymus, do the authors believe it is an altered education of developing thymocytes in CCR9^{-/-} mice?

The reviewer raises several thought-provoking points here. As discussed above, we have now evaluated whether CCL25 has an impact on CD4⁺CD8 $\alpha\alpha$ ⁺ IEL differentiation from peripheral CD4⁺ T cells. The proportion of CD4⁺CD8 $\alpha\alpha$ ⁺ IELs in *Ccl25*^{-/-} mice was comparable with that of *Ccl25*^{+/+} mice (littermate control) (R-Fig.1e). This data suggests that naïve CD4⁺ T cells in *Ccr9*^{-/-} mice have an inherently increased ability to differentiate into CD4⁺CD8 $\alpha\alpha$ ⁺ T cells, independent of CCL25-CCR9 interaction. Although the reviewer correctly notes that CCR9 is expressed in thymic T cells, we observed that splenic naïve CD4⁺ T cells from *Ccr9*^{-/-} mice expressed similar levels of *Runx3*, *Zbtb7b*, *Tbx21*, suggesting that naïve CD4⁺ T cells from *Ccr9*^{-/-} mice do not

highly express CD4-IEL-related genes as baseline (R-Fig. 4a, c). To further investigate this intriguing possibility, we cultured CD4⁺ T cells for 3 d under “IEL-differentiating conditions”⁶. Under these conditions, which mimicked the conditions encountered during lymph node to tissue migration, the percentage of CD4⁺CD8 $\alpha\alpha$ ⁺ T cells was higher in *Ccr9*^{-/-} mice than in *Ccr9*^{+/+} mice (R-Fig. 4a, b). Conversely, the expression of *Runx3* and *Tbx21* in cultured T cells from *Ccr9*^{-/-} mice was higher than that in *Ccr9*^{+/+} mice. However, the expression of *Zbtb7b* in cultured T cells was comparable between *Ccr9*^{-/-} and *Ccr9*^{+/+} mice (R-Fig. 4c).

We cultured naïve CD4⁺ T cells that were obtained from both *Ccr9*^{-/-} and *Ccr9*^{+/+} mice under IEL-differentiating conditions with the presence of CCL25. Our in vivo observations led us to expect that CCL25 would not have any effect on the percentage of CD4⁺CD8 $\alpha\alpha$ ⁺ T cells or the expression of *Runx3*, *Zbtb7b*, or *Tbx21* in *Ccr9*^{-/-} and *Ccr9*^{+/+} mice (R-Fig. 4b, c).

We also transferred the CD4⁺ T cells from ovalbumin (OVA)-specific T cell receptor transgenic (*Ccr9*^{+/+} OT-II) mice and *Ccr9*^{-/-} OT-II mice to *Rag2*^{-/-} mice. These mice were fed 1% ovalbumin for two weeks and analyzed. The proportion of CD4⁺CD8 $\alpha\alpha$ ⁺ T cells and Tregs in IELs was comparable between both groups. These data suggest that the deletion of the *Ccr9* gene does not have impact on the generation of CD4⁺CD8 $\alpha\alpha$ ⁺ T cells when TCR is identical (Reviewers only, R-Fig. 4d, e). We showed that the TCR diversity in spleen, MLN, and LPL cells was comparable between *Ccr9*^{+/+} mice and *Ccr9*^{-/-} mice, and the TCR diversity was reduced in IEL of *Ccr9*^{-/-} mice (Supplementary Fig. 6b). Nevertheless, as our group demonstrated, TCR engagement is essential for the induction of the CD4⁺CD8 $\alpha\alpha$ ⁺ T cells¹, and CD4⁺ T cell selection in *Ccr9*^{-/-} mice might work to promote the development of CD4⁺CD8 $\alpha\alpha$ ⁺ T cells.

We cannot rule out the possibility of altered thymic education because CCR9 is known as a chemokine receptor in thymic CD4⁺ T cells (and this possibility is now mentioned in the discussion). Nevertheless, our data suggest that *Ccr9*^{-/-} CD4⁺ T cells preferentially develop into CD4⁺CD8 $\alpha\alpha$ ⁺ T cells due to increased expression of *Runx3* and *Tbx21* when exposed to gut factors involved in IEL differentiation, regardless of whether CCL25 is present or not.

R-Fig. 4

(a) Scheme of experimental design. Naïve CD4⁺ T cells were obtained from *Ccr9*^{+/+} and *Ccr9*^{-/-} mice and cultured with anti-CD3/CD28, transforming growth factor-β (TGF-β), retinoic acid (RA), 2,3,7,8-tetrachlorodibenzodioxin (TCDD), and IFN-γ (IEL-differentiating condition), with or without CCL25. (b) Graph shows the frequency of CD8α⁺ cells among TCRβ⁺CD4⁺CD8β⁻ cultured cells of *Ccr9*^{+/+} and *Ccr9*^{-/-} mice. (c) Graphs show relative expression of *Runx3*, *Tbx21*, and *Zbtb7b* in naïve CD4⁺ T cells and cultured cells of *Ccr9*^{+/+} and *Ccr9*^{-/-} mice. (d) Scheme of experimental design. Naïve CD4⁺ T cells were obtained from *Ccr9*^{+/+} OT-II and *Ccr9*^{-/-} OT-II mice and transferred to *Rag2*^{-/-} mice. 1% Ovalbumin administration was started 1 d before the transfer, and 50 mg ovalbumin was administered by oral gavage on the day of transfer. 1% Ovalbumin was administered for two weeks before analysis. (e) Graphs show the frequency of CD8α⁺ or Foxp3⁺ populations among TCRβ⁺CD4⁺CD8β⁻ SI IELs. Statistical analysis was performed by one-way ANOVA, Tukey's post test (b, c), and Student's t-test (e). **P*<0.05, ***P*<0.01, ****P*<0.001, *****P*<0.0001.

The results of the mixed BM chimera experiments in F and G are highly surprising given that multiple studies have demonstrated that CCR9^{-/-}-BM is severely disadvantaged in generating peripheral CD4⁺ T cells in competitive settings with WT BM as the development of these cells is outcompeted within the thymus (see for example doi.org/10.1002/eji.200535203). Is this selective for the CD4⁺ IEL population and if so what are the authors explanation for this discrepancy?

We thank the reviewer for bringing up this important point. Indeed, Wurbel, M.A. demonstrated the reduction of T cells from *Ccr9*^{-/-} mice in the thymus⁷. In our set up after BMT, we evaluated the ratio of CD4⁺ T cells in spleen, MLN, and LPLs cells originating from *Ccr9*^{+/+} and *Ccr9*^{-/-} mice (R-Fig.5). We found that the ratio of CD4⁺ T cells in these cells was comparable between mouse lines. However, the proportion of CD4⁺CD8α⁺ IELs from *Ccr9*^{-/-} mice was higher than that from *Ccr9*^{+/+} mice (Fig. 2h, i, j). Our BMT results suggest that CD4⁺CD8α⁺ IEL differentiation is increased in *Ccr9*^{-/-} mice, but not due changes in homing per se. As discussed above, we now also demonstrate that CCR9-ligand CCL25 does not affect CD4⁺CD8α⁺ IEL differentiation. Although the precise mechanism of the competition between the different organs, such as the thymus and peripheral tissues, remains unclear, one possibility is that the reduced competitiveness of CCR9-deficient T cells is abrogated during the transition to the LP-IEL stage. This is likely due to their increased capacity for upregulating *Runx3* and Tbet, which promotes epithelium adaptation when compared to that in wild-type cells.

R-Fig. 5

Cells were obtained from bone marrow of *Cd45.1*⁺*Ccr9*^{+/+} and *Cd45.2*⁺*Ccr9*^{-/-} mice and mixed in a 1:1 ratio. Mixed bone marrow cells were transferred to lethally irradiated (11Gy) C57B6/J host mice and mice were analyzed four weeks after transfer. Graphs shows the percentage of CD45.1⁺ and CD45.2⁺ cells in TCRβ⁺CD4⁺CD8β⁻ SI LPLs, MLN cells, and splenocytes. Data are expressed as mean ± SD of individual mice (n=4, two pooled independent experiments). Statistical analysis was performed by Student's t test. **P*<0.05, ***P*<0.01, ****P*<0.001.

What is the authors explanation for the reduced numbers of FoxP3+CD4+ Tregs in the epithelium of CCR9-/- mice.

Our previous research demonstrated that Tregs in the LP compartment exhibit bidirectional movement to and from the IE compartment and that Tregs and CD4⁺CD8αα⁺ T cells have reciprocal roles in the IE compartment⁵. The total abundance and percentage of Tregs among LPLs was comparable between the two strains (Supplementary Fig 3a). While the precise mechanism of reduced Treg abundance in the IELs of *Ccr9*^{-/-} mice was unclear, we speculate that the increased number and percentage of CD4⁺CD8αα⁺ T cells in the IELs of *Ccr9*^{-/-} mice may occupy a niche in the intestinal epithelium, resulting in decreased number of Tregs among IELs.

Figure 3 and supplementary Figure 3.

As far as I can tell the scRNA-seq analysis was performed on one WT and one CCR9-/- mouse. If this is the case this should be repeated as this is insufficient to draw any firm conclusions on differences between these strains. Were these littermates or at a minimum co-housed?

We agree with the reviewer. While the original analysis was done using co-housed littermates, we have added two more mice (total 3 mice in each group) for scRNA-seq analysis and changed the figures accordingly.

The authors identified three CD4⁺ IEL clusters however it is unclear why they do not detect FOXP3⁺ Treg IEL identified in Figure 1 and 2. To this reviewer the major potential finding from this data is that Cluster 3 is completely absent from the IEL compartment of CCR9^{-/-} mice but for some reason this observation was not further explored (see below). What are these cells and can the authors confirm their absence in repeat scRNA-seq analysis or by flow cytometry?

In our previous study (original Fig. 3a-e), we identified Tregs as Cluster 6 in IELs. As mentioned above, we also added four more samples of IELs and LPLs from *Ccr9^{-/-}* and *Ccr9^{+/+}* mice and annotated a new cluster as the Treg population (R-Fig. 6a-d : Fig. 3a-e). Although original cluster 3 was reduced but not completely absent in *Ccr9^{-/-}* IELs, we were unable to find a specific surface marker for this cluster through differential gene expression analysis. It is possible that the differentiation of CD4⁺CD8α⁺ IELs involves a gradual program of multiple transcriptional and chromatin changes², which could explain why surface markers, except for CD8α, may not distinguish CD4⁺ IELs. Therefore, as the reviewer suggested, we performed scRNA-seq with two more mice in each group and identified 17 clusters in *Ccr9^{+/+}* and *Ccr9^{-/-}* mice. Through gene expression analysis, we determined that the new cluster 3 is equivalent to original cluster 3 (R-Fig. 6e). Furthermore, we found that the new cluster 3 was not completely absent in *Ccr9^{-/-}* IELs, and predominantly included of *Ccr9^{+/+}* IELs with some *Ccr9^{-/-}* IELs (R-Fig. 6d : Fig. 3e).

R-Fig. 6

Droplet-based single cell RNA sequencing (scRNA-seq) was performed using the Chromium 10X platform. CD4⁺ T cells from spleen, MLN, LPL, and IEL cells were sorted from *Ccr9*^{+/+} and *Ccr9*^{-/-} mice for scRNA-seq. (a) Cells were positioned by gene expression similarities, and 17 clusters were identified based on their top differentially expressed genes (DEGs), visualized by uniform manifold approximation and projection (UMAP). (b) UMAP representation of all sequenced cells, color-coded by cluster and separated by tissue of origin. (c) Heatmap of the top DEGs in each cluster. (d) Bar graph shows the proportion of cells in each cluster originating from spleen, MLN, LPL, and IEL cells of *Ccr9*^{+/+} and *Ccr9*^{-/-} mice. (e) Previous clusters annotate to new definitive clusters.

Figure 4 and supplementary Figure 4.

In Figure 4A, the resolution is so poor that it was not possible to see where clusters 0,1 and 3 lie in the trajectory. The authors state ‘The trajectory analysis indeed showed that Cluster 1 preceded Clusters 0 and 3 in CCR9+/+ mice and directly preceded Cluster 0 in CCR9-/- mice (Fig.4a), supporting that cells in cluster 1 were CD4+CD8αα+ IEL precursors’. To this reviewer this argument is circular as the authors must have set cluster 1 as their pseudo-time starting point or? If cluster 3 is clonally unrelated to cluster 1 and 0 what is the authors rational for including this population in the trajectory analysis? It has also been suggest that FOXP3+ Tregs are the immediate precursor of at least a proportion of CD4+CD8αα+ IEL with regulatory functon(DOI: 10.1126/science.aaf3892). Do such cells connect with CD4+CD8αα+ IEL in trajectory analysis?

We apologize for this oversight and thank the reviewer for raising these points.

Additional samples have been added and the figures have been updated. Our analysis began with the naïve CD4⁺ T cell population (cluster 6), and through pseudotime analysis, we observed that these cells differentiated into CD4⁺CD8αα⁺ T cells (R-Fig.7a : Fig. 4a). This differentiation process involved Tregs (Cluster 11), as we have previously demonstrated that some ex-Tregs give rise to CD4⁺CD8αα⁺ T cells. Our previous research has shown that Tregs share a similar developmental program with CD4⁺CD8αα⁺ IELs based on fate-mapping and chromatin analysis^{2, 5}. These findings, taken together, support the link between Tregs in IELs and CD4⁺CD8αα⁺ IELs in both *Ccr9*^{+/+} and *Ccr9*^{-/-} mice.

R-Fig. 7

(a) Pseudotime analysis of sequenced CD4⁺ T cells originating from *Ccr9*^{+/+} (left) and *Ccr9*^{-/-} mice (right), color-coded by pseudotime gradient. Numbers on the UMAP figure indicate the cluster numbers. (b) Violin plots show the expression of *Runx3*, *Tbx21*, *Cbfb*, *Zbtb7b*, and *CD8a* among IELs of clusters 0, 4, and 7 (CD4⁺CD8α⁺ T cells); clusters 1, 2, and 3 (CD4⁺CD8α⁻ T cells); and clusters 10, 13, and 14 (CD4⁺CD8α^{int} T cells). (c) Violin plots show the expression of *Runx3*, *Tbx21*, and *Cbfb* among IELs of clusters 0, 4, and 7 (CD4⁺CD8α⁺ T cells); clusters 1, 2, and 3 (CD4⁺CD8α⁻ T cells); and clusters 10, 13, and 14 (CD4⁺CD8α^{int} T cells) from *Ccr9*^{+/+} and *Ccr9*^{-/-} mice. Statistical analysis was performed by one-way ANOVA, Tukey's post test (b), and Mann-Whitney-Wilcoxon test (c). **P*<0.05, ***P*<0.01, ****P*<0.001.

For comparing gene between clusters and between WT and CCR9^{-/-} mice it is not appropriate to perform statistical analysis between one WT and one CCR9^{-/-} mouse (Figure 4C, Figure S4I). For the PCR analysis in Figure 4D, the enhanced expression of *Runx3*, *Tbx21* and *Cbfp2* in CCR9^{-/-} CD4⁺CD8α⁺ and CD4⁺CD8α⁻ IEL could simply be that these mice lack cluster 3 cells which express reduced levels of these markers? Indeed there appears to be no difference in *Cbfb* expression when comparing within the same cluster between CCR9^{-/-} and WT mice.

Thank you for your comments. We have incorporated two additional samples, bringing the total to three in each group, and reanalyzed the expression of *Runx3*, *Tbx21* and *Cbfb* in each IEL cluster, both in *Ccr9*^{-/-} and *Ccr9*^{+/+} mice. Our findings indicate that the CD4⁺CD8α⁺ IEL clusters (clusters 0, 4, and 7) exhibited higher expression of both *Runx3* and *Tbx21*, than the CD4⁺CD8α⁻ IEL clusters (clusters 1, 2, and 3) and the CD4⁺CD8α^{int} IEL clusters (clusters 10, 13, and 14) (R-Fig. 7b : Fig. 4b : Supplemental table 2). The CD4⁺CD8α⁻ IEL clusters (cluster 1, 2, and 3) and the CD4⁺CD8α⁺ IEL clusters (cluster 0, 4, and 7) in *Ccr9*^{-/-} mice exhibited a higher level of *Runx3*, *Tbx21*, and *Cbfb* expression than those in *Ccr9*^{+/+} mice (R-Fig.7c : Fig. 4c). CD4⁺CD8α^{int} IEL clusters (cluster 13 and 14) in *Ccr9*^{-/-} mice also displayed a tendency towards higher expression of *Runx3*, *Tbx21*, and *Cbfb* than those in *Ccr9*^{+/+} mice. These scRNA-seq data align with our PCR data, supporting the notion that enhanced expression of *Cbfb*, *Runx3*, and *Tbx21* in CD4⁺CD8α⁻ T cells of *Ccr9*^{-/-} mice strengthens their capacity to become CD4⁺CD8α⁺ T cells.

Figure 6.

While the results regarding Runx3, Cbfb and Tbet knockdown are somewhat expected, the result that knockdown of CCR9 enhances the ability of CD4⁺ T cells to convert into CD4⁺CD8 $\alpha\alpha$ ⁻ IEL-like cells is very interesting but not explored. What are the underlying mechanisms to this observation, in particular as the ligand CCL25 appears to be absent. What happens when CCL25 is added to the in vitro cocktail? What impact does CCL25 have on the transcriptional profile of these cells and are there differences in the transcriptional profile of CCR9^{-/-} and WT CD4⁺ T cells at the start and after 3 days of culture. This system would appear ideal for identifying the potential mechanisms driving CD4⁺ IEL alterations in CCR9^{-/-} mice.

We demonstrated that CCL25 was not involved in the CD4⁺CD8 $\alpha\alpha$ ⁺ T cell development in vivo or in vitro (R-Fig 4. discussed above).

Minor points

The authors write ‘Collectively, these results showed that upregulation of Cbfb β 2 in CD4⁺ T cells is essential for CD4⁺CD8 $\alpha\alpha$ ⁺ IELs differentiation’. Please tone this down as CD4⁺CD8 $\alpha\alpha$ ⁺ IELs are clearly present in Cbfb β 2 deficient mice.

Thank you for your valuable suggestion. Indeed, CD4⁺CD8 $\alpha\alpha$ ⁺ T cells were found in Cbfb β 2 deficient mice. We toned down this sentence as follows: “Collectively, these results showed that upregulation of Cbfb β 2 in CD4⁺ T cells facilitated their differentiation towards CD4⁺CD8 $\alpha\alpha$ ⁺ T cells”.

Similarly, in the results heading the authors write ‘Downregulation of CCR9...’ and ‘CCR9 downregulation...’ when comparing CCR9^{-/-} to WT mice and these titles should be changed to more accurately reflect what they are looking at.

We changed the language used to compare *Ccr9*^{-/-} and *Ccr9*^{+/+} mice.

Reviewer #2 (Remarks to the Author):

Major Concerns:

1. It is unclear why the gating of CCR9 differs from IELs and LPLs in Fig 1a and between spleen and MLN in Extended Figure 1b. Adding an FMO or isotype control would provide more confidence in the analysis and interpretation of the data presented.

The reason for the difference in FACS gating between LPLs and IELs was due to the use of collagenase to obtain LPLs, which can reduce the expression of various surface proteins. In response to the reviewer's suggestion, we have now included FMO analysis to display CCR9 expression in each population. We also confirmed that the CCR9 expression in CD4⁺CD8 α ⁻ T cells and Tregs was higher than that in CD4⁺CD8 α ⁺ T cells both in IELs and LPLs (R-Fig. 1c, d; Fig. 1a, b).

2. CD4⁺ CD8 α ⁺ T cells develop in the epithelium and not LP in WT mice but are found in both locations in CCR9 KO mice. The authors conclude that the “developmental process does not start until the lymphocytes enter the epithelium”. First, it should be clarified that the prior statement applies to WT mice. Also, can the authors provide an explanation for why there are DP T cells in the LP in the CCR9 KO? Is there a failure of CCR9 KO DP LPLs to migrate into the epithelium? Are there DEGs between the DP IELs and LPLs from CCR9 KO mice that could provide some insight?

The reviewer raises here several important points, some of which were not addressed in our original submission. We performed several experiments to address these points, as follows:

- We confirmed the existence of CD4⁺CD8 α ⁺ T cells in the LP of *Ccr9*^{-/-} mice by staining intestinal tissue for CD4 and CD8 α (R-Fig. 8).

- Next, we manipulated the expression of *Zbtb7b* and *Runx3* in CD4⁺ T cells⁴ and observed the appearance of CD4⁺CD8αα⁺ T cells in both the LP and mesenteric lymph nodes (MLNs). These results suggest that the parallel manipulation of factors facilitates CD4⁺CD8αα⁺ IEL differentiation.
- Thpok downregulation and Runx3 upregulation could induce IEL programming in CD4⁺ T cells before they reach the epithelium. Moreover, we also compared the gene expression profiles of CD4⁺CD8αα⁺ T cells in the IE compartment and LP of *Ccr9*^{-/-} mice (R-Fig. 9a, b). Our findings reveal that genes involved in CD4⁺CD8αα⁺ IEL development, including *Runx3*, *Itgae*, and *Cd8a*, were expressed in CD4⁺CD8αα⁺ T cells in both the IE compartment and the LP (refer to Figure 9b). However, the expression of other genes, such as *Zeb2* (Cytotoxic T cell promotion gene with Tbx21)⁸ and *Etv6* (ETS family gene to promote Cbfβ-Runx3 interaction)⁹ in IE compartment was higher than in the LP. We have previously demonstrated that IELs migrate between the IE compartment and the LP before settling in the IE compartment. Thus, our data suggest that CD4⁺CD8αα⁺ T cells in *Ccr9*^{-/-} mice in LP are undergoing development towards becoming IELs.

R-Fig. 8

Representative small intestine vibratome section obtained from *Ccr9*^{+/+} and *Ccr9*^{-/-} mice. Fixed small intestine samples were stained for CD4, CD8α, and TCRβ with actin, and then imaged with a confocal, high-content imaging system.

d Expression of *Mki67* in CD4⁺CD8 α ⁺ (cluster 1,2,3)

Expression of *Mki67* in CD4⁺CD8 α ⁺ (cluster 0,4,7)

R-Fig. 9

(a) Volcano plot of differentially expressed genes among clusters 0, 4, and 7 between *Ccr9*^{-/-} IELs and LPLs (P<0.05, in color). Upregulated and downregulated genes in *Ccr9*^{-/-} IELs are shown as red and blue points, respectively. Genes with no difference are shown as black points. (b) Expression of *Cd4*, *Cd8a*, *Runx3*, *Tbx21*, *Cbfb*, and *Itgae* genes among *Ccr9*^{-/-} IELs and LPLs. (c) Expression of *Mki67* in IELs and LPLs of *Ccr9*^{+/+} and *Ccr9*^{-/-} mice are shown on a UMAP figure. (d) Expression of *Mki67* in CD4⁺CD8α⁻ T cells (clusters 1, 2, and 3) and CD4⁺CD8α⁺ T cells (cluster 0, 4, 7) among IELs and LPLs of *Ccr9*^{+/+} and *Ccr9*^{-/-} mice.

3. Previous reports have characterized CCR9 downstream signaling in lymphocytes. While determining the exact mechanism restrains the DP IEL developmental program is beyond the scope of this study, it would be useful to discuss potential CCR9 downstream signals that may be involved.

CCR9-mediated signals activate PI3K/AKT in solid cancer; however, the role of CCR9 receptor signal in lymphoid cells besides attracting them to CCL25 was not known¹⁰⁻¹². We cultured naïve CD4⁺ T cells both from *Ccr9*^{+/+} and *Ccr9*^{-/-} mice with or without CCL25 under IEL-differentiating conditions. The percentage of CD4⁺CD8α⁺ T cells was similar between samples, regardless of the presence of CCL25 (R-Fig. 4a, b). Moreover, the percentage of CD4⁺CD8α⁺ IELs was comparable between *Ccl25*^{-/-} and *Ccl25*^{+/+} mice (R-Fig. 1e). It is unclear whether CCR9 signaling is induced by CCL25 alone, but CCL25-CCR9 signaling was not involved in the development of the CD4⁺CD8α⁺ T cells *in vitro* or *in vivo*. Downregulation of CCR9 in CD4⁺ T cells may represent a potential approach to modify the PI3K/AKT signal and upregulate *Cbfb*, *Runx3*, and *Tbx21*.

Minor Concerns:

1. Page 6, the authors show that vitA, but not the microbiota is required for CCR9 expression; however, the conclusion states that “like microbiota, enhanced CD4+ CD8aa+ IEL differentiation in CCR9 KO mice is RA-dependent.” The way this is worded is confusing because the data show that the microbiota is not required for the phenotype observed in CCR9 KO mice.

Because the gut microbiota is essential for the differentiation of CD4⁺CD8 $\alpha\alpha$ ⁺ IELs, even in *Ccr9*^{-/-} mice, we have changed the text accordingly to clarify this point.

2. The authors demonstrate that the frequency of IFN γ -expressing cells is decreased in the CD4-Cbfb2Tg mice and conclude that there are decreased Th1 LPLs, yet this point is not discussed further in the discussion.

Indeed, Tbet expression is indispensable for the differentiation of CD4⁺CD8 $\alpha\alpha$ ⁺ T cells⁶; however, its connection to IFN γ and Cbfb β 2 is not known. We now acknowledge this point in the discussion, as suggested.

Reviewer #3 (Remarks to the Author):

The authors investigated the effects of CCR9 expression and T cell differentiation. Overall, the manuscript was very difficult to read due to the wording of sentences and changing of subjects between connecting sentences. Additionally, it was not well defined why CCR9 is of importance.

For instance in the abstract, the second and third sentence seem to contradict one another. It is mentioned T cell differentiation can develop at the epithelium by the lamina propria and expression of CCR9 is low. The problem of interest is to define the mechanism preventing differentiation in LP but then later states that low CCR9 expression is found in epithelial IELS and results in over differentiation in both epithelium and LP. How does CCR9 prevent differentiation in LP if it is shown to have over differentiation?

Thank you for your comments. We carefully and comprehensively rewrote the manuscript to improve clarity and flow.

The reviewer makes several important and interesting points that, unfortunately, our original version did not clarify well. Indeed, the original text did not clearly address the relationship between CCR9 expression and IEL differentiation. The revised manuscript explains that the overexpression of CCR9 does not prevent CD4⁺CD8 $\alpha\alpha$ ⁺ IEL

differentiation, although the lack of CCR9 expression leads to an increased percentage of CD4⁺CD8 $\alpha\alpha$ ⁺ IELs. We interpreted this apparently contrasting data as being related to the signaling function of CCR9. During physiological differentiation, CCR9 downregulation facilitates the induction of a T cell-epithelium program, which is mimicked by CCR9 deletion. Overexpression of CCR9, in contrast, is not sufficient to prevent this program induction once T cells arise in the epithelium.

Previously, we observed epigenetic and chromatin modification during the development of IELs². A lack of CCR9 might enhance these epigenetic and chromatin modifications and facilitate the adaptation of CD4⁺CD8 $\alpha\alpha$ ⁺ IELs with increased expression of CD4⁺CD8 $\alpha\alpha$ ⁺ IEL-related genes, such as *Runx3*, *Cbfb2*, and *Tbx21*. However, it appeared that the overexpression of CCR9 alone was not adequate to induce the necessary epigenetic modifications for CD4⁺CD8 $\alpha\alpha$ ⁺ IEL differentiation. Moreover, it is possible that CCR9 overexpression in the absence of signaling partners, normally changed in epithelial T cells, is not sufficient to interfere with this program.

It was previously reported that CCR9 is expressed in thymic T cells, so specific TCR on T cells in *Ccr9*^{-/-} mice might facilitate the development of CD4⁺CD8 $\alpha\alpha$ ⁺ IELs. Given that the homing capacity to the thymus is reduced by CCR9 deficiency, it is possible that CCR9-deficient thymic T cells undergo a distinct primal differentiation process in the thymus to compensate for the lower number of pre-thymic T cell progenitors. As TCR signaling engagement is essential for the induction of the CD4⁺CD8 $\alpha\alpha$ ⁺ T cells¹, a third possibility is alterations to the TCR repertoire could lead to an enrichment of CD4⁺ T cell pools that are more likely to differentiate into CD4⁺CD8 $\alpha\alpha$ ⁺ IELs; this might not occur in CCR9 transgenic mice. Our TCR analysis suggests that the acquisition of CD4⁺CD8 $\alpha\alpha$ ⁺ IEL characteristics due to CCR9 deficiency is unlikely caused by a biased TCR repertoire; nevertheless, further analysis is needed to uncover the underlying mechanism.

We have added these possibilities to the discussion. Furthermore, the revised version highlights the fact that a lack of CCR9 expression increased the relative abundance of CD4⁺CD8 $\alpha\alpha$ ⁺ IELs *in vivo* and *in vitro*, likely through increased expression of Cbfb β 2. Overall, the revised manuscript provides a clearer and more coherent explanation of the study's findings, addressing the points raised by the reviewer and improving the overall quality of the manuscript.

1. Third paragraph of the introduction: The first two sentences describe the expression of different subjects (gene or cells) making it difficult to understand. The first sentence describes CCR9 expression is down-regulated in CD4+ differentiation but the next sentence switches to IELs differentiation is increased in a different cell type, CD4+CD8+.

Thank you for the valuable comments. We have changed the text as follows:

“In the present study, we discovered that CCR9 expression is downregulated in CD4⁺CD8αα⁺ IELs compared to that in CD4⁺CD8αα⁻ IELs and Tregs in the IE compartment. Conversely, CD4⁺CD8αα⁺ IELs were more abundant in *Ccr9*^{-/-} mice than in *Ccr9*^{+/+} mice. However, we also found that the CCR9 ligand, CCL25, did not interfere with CD4⁺CD8αα⁺ IEL differentiation. Single-cell RNA sequencing (scRNA-seq) of IELs from *Ccr9*^{-/-} and *Ccr9*^{+/+} mice revealed that CD4⁺ T cells in the epithelia of *Ccr9*^{-/-} mice exhibited greater expression of *Runx3*, *Tbx21*, and *Cbfb*, which are essential for the development of CD4⁺CD8αα⁺ IELs, than those of *Ccr9*^{+/+} mice. *Cbfb* has two splicing forms, *Cbfb1* and *Cbfb2*, and we observed that *Cbfb2*, but not *Cbfb1*, was more highly expressed in CD4⁺ IELs from *Ccr9*^{-/-} mice than in those from *Ccr9*^{+/+} mice. Additionally, transgenic expression of *Cbfb2* increased the differentiation of CD4⁺CD8αα⁺ IELs *in vivo*. In summary, our findings suggest that CCR9 downregulation in CD4⁺ T cells facilitates their differentiation towards epithelium-adapted CD4⁺CD8αα⁺ IELs via upregulation of the Runx3-partner *Cbfb2*.”

We have edited the entire text to improve flow.

2. In the 'Loss of CCR9 enhances CD4+CD8+ IEL differentiation' section, paragraph 4: “Thus while downregulation of CCR9 strongly facilitates CD4+CD8+ cell differentiation, CCR9 overexpression is not sufficient to prevent this process.

The overexpression of CCR9 does not decrease CD4+CD8+ IELs differentiation. What is the importance in studying CCR9?

Although CCR9 is known as a gut homing receptor, its possible role in cell differentiation remains unclear. Indeed, in contrast to the data showing physiological downregulation of CCR9 during T cell adaptation to the gut epithelium, and in line with the CCR9 knockout data, we observed that the main CCR9 chemokine ligand, CCL25, did not affect the development of CD4⁺CD8 $\alpha\alpha$ ⁺ T cell differentiation *in vitro* or *in vivo*. Therefore, although CCR9-CCL25 signaling does not seem to be involved in CD4⁺CD8 $\alpha\alpha$ ⁺ T cell differentiation, CCR9 deficiency results in an increased abundance of CD4⁺CD8 $\alpha\alpha$ ⁺ T cells. We have raised several possibilities for this observation, including a lack of epigenetic and chromatin modification, signaling partners to become CD4⁺CD8 $\alpha\alpha$ ⁺ IELs, or TCR engagement.

We speculate that the chemokine receptor CCR9 plays a role not only in immune cell attraction but also cell development in CD4⁺CD8 $\alpha\alpha$ ⁺ T cells. We have added these possibilities to the discussion. The precise mechanism underlying CCR9 signal involvement of CD4⁺CD8 $\alpha\alpha$ ⁺ T cell differentiation is unclear, but we believe that the involvement of a chemokine receptor in cell development is important.

3. A model figure of the cell differentiation process and the experimental design would be extremely helpful.

Thank you for the valuable comments. We added an experimental design schema to improve clarity (R-Fig. 4a : Fig. 2d : Supplementary Fig. 3l).

4. In the single-cell RNA experiment, the pseudotime analysis is not described in the text. The statement “ These data support that CD4⁺CD8⁺ T cells development starts in the intraepithelial compartment of Ccr9^{-/-} mice “ is incorrect with only the cell type analysis. This is stated for Figure 3.

Thank you for the comment. As the reviewer correctly points out, these data did not indicate where CD4⁺CD8 $\alpha\alpha$ ⁺ T cells start to develop. We removed the statement above from Figure 3. We have included a description of pseudotime analysis in the text (related to Figure 4). The pseudotime analysis revealed that CD4⁺CD8 $\alpha\alpha$ ⁺ T cells (clusters 0, 4, and 7) were derived from CD4⁺CD8 $\alpha\alpha$ ⁻ T cells (clusters 1, 2, and 3).

Some CD4⁺ T cells from *Ccr9*^{+/+} mice developed and stopped at clusters 1, 2, and 3, while CD4⁺ T cells from *Ccr9*^{-/-} mice passed through clusters 1–3 and differentiated into CD4⁺CD8αα⁺ IELs. Taken together, these results show that the development of CD4⁺ T cells from *Ccr9*^{+/+} mice and *Ccr9*^{-/-} mice differed, resulting in a lower frequency of clusters 1, 2, and 3 from *Ccr9*^{-/-} mice and a higher frequency of clusters 0, 4, and 7.

5. What are the cell number counts per tissue type? Were more cells sequenced for a particular type?

With the inclusion of more mice, we sequenced 7,868 IELs, 3,546 LPLs, 448 MLNs, and 453 splenocytes of *Ccr9*^{+/+} mice and 8,434 IELs, 3,204 LPLs, 338 MLNs, and 216 splenocytes of *Ccr9*^{-/-} mice.

6. Figure 4A is not described in the main body of the text.

We apologize for this oversight. We have now corrected this mistake in the main text.

7. The scRNAseq data shows CD4+CD8+ IELS in *Ccr9*^{-/-} and *Ccr9*^{+/+} mice to be identical. This should be the expected result since the only change is to knock out *Ccr9*. The next statement, “Overall, the above data suggest that clonal expansion of CD4+CD8- IEL precursors in *Ccr9*^{-/-} mice likely takes place after migration to the IE compartment”, switches from CD4+CD8+ to CD4+CD8- cells. Also, how does this data suggest where clonal expansion occurs? Is this part of the pseudotime analysis that is not mentioned?

We thank the reviewer for pointing this out. The reviewer is correct, and we agree that the term “clonal expansion” was inappropriately used here.

First, we analyzed the expression of *Mki67* in CD4⁺CD8 α ⁻ T cell clusters (clusters 1, 2, and 3) and CD4⁺CD8 α ⁺ clusters (clusters 0, 4, 7) from IELs and LPLs of each strain. In CD4⁺CD8 α ⁻ and CD4⁺CD8 α ⁺ clusters, the IEL population from *Ccr9*^{-/-} mice was larger than other populations, such as LPLs from *Ccr9*^{-/-} mice or IELs from *Ccr9*^{+/+} mice. These data suggest that *Mki67* is highly expressed in all IEL clusters from *Ccr9*^{-/-} mice (R-Fig. 9c, d, for reviewer only). We then analyzed the expression of *Ki67* and *EdU* in CD4⁺CD8 α ⁻ and CD4⁺CD8 α ⁺ IELs. While their expression in CD4⁺CD8 α ⁻ IELs was comparable between *Ccr9*^{-/-} and *Ccr9*^{+/+} mice, their expression in CD4⁺CD8 α ⁺ IELs of *Ccr9*^{-/-} mice was higher than that in *Ccr9*^{+/+} mice. Taken together, our data suggest CD4⁺CD8 α ⁺ IELs of *Ccr9*^{-/-} mice proliferate more than those of *Ccr9*^{+/+} mice. To show which cluster the cells are enriched by monoclonal cells, we analyzed the TCR repertoire and determined the top 5 TCR clonotype cells in IELs from both strains. Indeed, although the top 5 TCR clonotype cells were distributed in all IEL clusters (CD4⁺CD8 α ⁻ and CD4⁺CD8 α ⁺ cells) from *Ccr9*^{+/+} mice, the top 5 TCR clonotype cells were preferentially distributed in CD4⁺CD8 α ⁺ clusters from *Ccr9*^{-/-} mice (Supplementary fig. 6d). Hence, our data suggest that CD4⁺CD8 α ⁺ clusters from *Ccr9* deficient mice may continue proliferating upon LP-to-IE compartment migration, unlike their wild-type counterparts.

8. Discussion. “ Our results indicate that CCR9 loss functionally facilitates CD4+ T cell precursors to acquire an IEL program..” How was this shown when overexpression also shows differentiation?

Thank you for the constructive comment. Although CCR9 is known as a gut homing receptor, its possible role in cell differentiation remains unclear. We have raised several possibilities for this above, including epigenetic and chromatin modifications, the signaling partners promoting CD4⁺CD8 α ⁺ IELs, and the TCR engagement. We speculate that the chemokine receptor CCR9 plays a role not only in immune cell attraction but also the development of CD4⁺CD8 α ⁺ T cells. We have added these possibilities to the discussion.

9. The figures are small and hard to read.

We changed the figures to be more reader-friendly.

10. Data availability statement: Is there a dataset identifier from the DNA Data Bank of Japan?

We added the dataset identifier to the methods section.

References

1. Bilate, A. M. et al. T cell receptor is required for differentiation, but not maintenance, of intestinal CD4⁺ intraepithelial lymphocytes. *Immunity* **53**, 1001-1014 e1020 (2020). [10.1016/j.immuni.2020.09.003](https://doi.org/10.1016/j.immuni.2020.09.003)
2. London, M., Bilate, A. M., Castro, T. B. R., Sujino, T. & Mucida, D. Stepwise chromatin and transcriptional acquisition of an intraepithelial lymphocyte program. *Nat. Immunol.* **22**, 449-459 (2021). [10.1038/s41590-021-00883-8](https://doi.org/10.1038/s41590-021-00883-8)
3. Mucida, D. et al. Transcriptional reprogramming of mature CD4⁺ helper T cells generates distinct MHC class II-restricted cytotoxic T lymphocytes. *Nat. Immunol.* **14**, 281-289 (2013). [10.1038/ni.2523](https://doi.org/10.1038/ni.2523)
4. Reis, B. S., Rogoz, A., Costa-Pinto, F. A., Taniuchi, I. & Mucida, D. Mutual expression of the transcription factors Runx3 and ThPOK regulates intestinal CD4⁺ T cell immunity. *Nat. Immunol.* **14**, 271-280 (2013). [10.1038/ni.2518](https://doi.org/10.1038/ni.2518)
5. Sujino, T. et al. Tissue adaptation of regulatory and intraepithelial CD4⁺ T cells controls gut inflammation. *Science* **352**, 1581–1586 (2016). [10.1126/science.aaf3892](https://doi.org/10.1126/science.aaf3892)
6. Reis, B. S., Hoytema van Konijnenburg, D. P., Grivennikov, S. I. & Mucida, D. Transcription factor T-bet regulates intraepithelial lymphocyte functional maturation. *Immunity* **41**, 244-256 (2014). [10.1016/j.immuni.2014.06.017](https://doi.org/10.1016/j.immuni.2014.06.017)
7. Wurbel, M. A., Malissen, B. & Campbell, J. J. Complex regulation of CCR9 at multiple discrete stages of T cell development. *Eur. J. Immunol.* **36**, 73–81 (2006). [10.1002/eji.200535203](https://doi.org/10.1002/eji.200535203)

8. Dominguez, C. X. et al. The transcription factors ZEB2 and T-bet cooperate to program cytotoxic T cell terminal differentiation in response to LCMV viral infection. *J. Exp. Med.* **212**, 2041-2056 (2015). [10.1084/jem.20150186](https://doi.org/10.1084/jem.20150186)
9. Gu, T. L., Goetz, T. L., Graves, B. J. & Speck, N. A. Auto-inhibition and partner proteins, core-binding factor β (CBF β) and Ets-1, modulate DNA binding by CBF α 2 (AML1). *Mol. Cell. Biol.* **20**, 91–103 (2000). [10.1128/MCB.20.1.91-103.2000](https://doi.org/10.1128/MCB.20.1.91-103.2000)
10. Lee, S. et al. CCR9-mediated signaling through beta-catenin and identification of a novel CCR9 antagonist. *Mol. Oncol.* **9**, 1599-1611 (2015). [10.1016/j.molonc.2015.04.012](https://doi.org/10.1016/j.molonc.2015.04.012)
11. Deng, X. et al. Wnt5a and CCL25 promote adult T-cell acute lymphoblastic leukemia cell migration, invasion and metastasis. *Oncotarget* **8**, 39033-39047 (2017). [10.18632/oncotarget.16559](https://doi.org/10.18632/oncotarget.16559)
12. Wang, C. et al. The role of chemokine receptor 9/chemokine ligand 25 signaling: From immune cells to cancer cells. *Onco.l Lett.* **16**, 2071-2077 (2018). [10.3892/ol.2018.8896](https://doi.org/10.3892/ol.2018.8896)

REVIEWER COMMENTS

Reviewer #1 (Remarks to the Author):

While the authors have made efforts to address my comments, I feel that some of my original concerns have not been addressed sufficiently and I have additional concerns on the new data. Importantly, the mechanism underlying the observed CCR9 dependent effects (that now appear CCL25 independent) remain completely unclear. The language in many places remains poor and difficult to follow.

Regarding my Figure 1 query, the authors have now examined CCL25 KO mice but they do not directly address my question regarding the possibility of ligand induced regulation of CCR9 levels on CD4+ IEL subsets. The authors should examine and show data on CCR9 expression on IEL subsets comparing WT with CCL25 KO littermates to determine whether CCL25 mediated CCR9 regulation could underlie the differences in CCR9 expression on IEL as they enter and are maintained within the epithelium. This is essential for interpretation of the results where the authors are suggesting CCL25 independent regulation of CCR9 expression as a modulator of CD4+CD8 $\alpha\alpha$ + IEL development.

The authors do not provide an explanation as to why their mixed CCR9^{-/-}/WT BM chimera results in Fig 4G and H differ so dramatically from multiple published CCR9^{-/-}/WT mixed BM chimera studies (not just doi.org/10.1002/eji.200535203) that demonstrate that CCR9^{-/-} BM is competitively disadvantaged in generating T cells compared with WT BM. This is because CCR9^{-/-} cells are outcompeted by WT cells in the thymus.

In Figure 3 I am glad to see in the revised manuscript that the authors have repeated the scRNA-seq data and not based their conclusions on comparisons between 1 mouse in each group. Nevertheless, I have several issues with the presentation of the new data. From the presented UMAPs it is difficult to assess alterations in subset proportions (1) between different sites, (2) between CCR9 KO and WT mice and (3) between biological replicates. Instead of the UMAPs in Figure 3B the authors should show bar graphs with proportion of each subset within each of the 4 sites and for each of the individual 6 mice to show how reproducible the data is. The definition of the subsets is also very unclear. What genes were

used to define each subset? Why for example are there Tfh cells in the IEL and LPL preps where they would not be expected, and not in the spleen or MLN where they would be expected? Again, in Fig 3D it is impossible to decipher the proportions of each subset between CCR9 KO and WT cells and it is not possible to interpret the data in Fig 3E without the same exact cell numbers being analysed from each site and strain of mouse. The authors need to show cell numbers analysed for each sample and strain. Please present and compare such data in bar graphs with proportions of each subset within a given tissue and mouse and then comparing the three CCR9 KO vs three WT mice but also showing a bar graph for each individual mouse to show reproducibility.

Based on the results in Figure 4 the authors make several strong conclusions regarding cellular trajectories and development of CD4+CD8 $\alpha\alpha$ + IEL and alterations in KO mice. The conclusions however are based purely on bioinformatic analysis using one trajectory analysis algorithm without direct evidence. The authors should acknowledge this and tone down there conclusions accordingly.

Minor point

How many mice are presented in Supplementary Figure 6C and were similar observations made for each mouse?

Reviewer #2 (Remarks to the Author):

The authors have largely addressed my concerns from the prior review. However, in the revision (Supp Fig 3n), the authors state that “cultured T cells from CCR9 KO mice exhibited higher Runx3 and Tbx21 expression than those from CCR9 WT mice.” In this figure, there is a significant reduction in Tbx21 expression between naïve and cultured T cells within the same genotype. No differences between the two genotypes are shown, and thus is there no support for this statement. It is possible that the ex vivo culture conditions may not accurately reflect what the in vivo biology reflected in the scRNAseq data (Fig 5) that does support the authors’ overall conclusion.

In Figure 1a, it appears that there are multiple colored lines behind the FMO in the IEL

histograms. These are not in the LPL histograms and may be included in error, otherwise, it is unclear what data those additional lines represent.

Reviewer #3 (Remarks to the Author):

Thank you for addressing my concerns and providing more clarity and data to the manuscript. I have no more concerns.

We would like to express our gratitude to the reviewers for their feedback on our manuscript. We have carefully considered the points raised by the reviewers and have made the following revisions

1. We have added figures related to CCR9 expression in the IELs of *Ccl25*^{-/-} mice. (Reviewer #1).
2. As we mislabeled the *Tbx21* expression in the Figure, we have corrected the Figure (Reviewer #2).
3. We have provided further clarification regarding the single-cell RNA-seq experiments. We have added the proportion of each cluster from the single-cell RNA-seq data to each mouse and showed the biological replicates (Reviewer #1).
4. We have modified the figures and manuscript according to the reviewers' suggestions (Reviewer #1, 2).
5. We add the 2 more mice to the BMT experiments and re-analyzed the data, including the spleen. Similar to previous reports, we observed that *Ccr9*^{+/+} T cells competed with *Ccr9*^{-/-} T cells in the spleen, mLN, and IELs. Nevertheless, the percentage of CD4⁺CD8 α ⁺ IELs in *Ccr9*^{-/-} mice was higher than in *Ccr9*^{+/+} mice (Reviewer #1).

REVIEWER COMMENTS

Reviewer #1 (Remarks to the Author):

While the authors have made efforts to address my comments, I feel that some of my original concerns have not been addressed sufficiently and I have additional concerns on the new data. Importantly, the mechanism underlying the observed CCR9 dependent effects (that now appear CCL25 independent) remain completely unclear. The language in many places remains poor and difficult to follow.

Regarding my Figure 1 query, the authors have now examined CCL25 KO mice but they do not directly address my question regarding the possibility of ligand induced regulation of CCR9 levels on CD4⁺ IEL subsets. The authors should examine and show data on CCR9 expression on IEL subsets comparing WT with CCL25 KO littermates to determine whether CCL25

mediated CCR9 regulation could underlie the differences in CCR9 expression on IEL as they enter and are maintained within the epithelium. This is essential for interpretation of the results where the authors are suggesting CCL25 independent regulation of CCR9 expression as a modulator of CD4⁺CD8⁺ IEL development.

Thank you for your comments. We agree with your suggestion and have performed experiments analyzing CCR9 expression in *Ccl25*^{-/-} and *Ccl25*^{+/+} mice (R-Fig. 1a, b). Indeed, as the reviewer pointed out, there seems to be CCL25-dependent regulation of CCR9 expression. CCR9 expression in Tregs, CD4⁺CD8 α ⁻ T cells, and CD4⁺CD8 α ⁺ T cells in IELs of *Ccl25*^{-/-} mice was upregulated higher compared to that in *Ccl25*^{+/+} mice (R-Fig. 1a, b). However, similar to our observations in WT mice, we found that, in *Ccl25*^{-/-} mice, the MFI of CCR9 in Tregs from *Ccl25*^{-/-} mice was higher than that of CD4⁺CD8 α ⁻ IELs or CD4⁺CD8 α ⁺ IELs from *Ccl25*^{-/-} mice (R-Fig. 1b). Similarly, the MFI of CCR9 in CD4⁺CD8 α ⁻ IELs from *Ccl25*^{-/-} mice was higher than that in CD4⁺CD8 α ⁺ IELs from *Ccl25*^{-/-} mice. Hence, while the expression level of CCR9 was high in *Ccl25*^{-/-} mice due to the lack of CCL25; however, the patterns of CCR9 expression observed in WT mice were recapitulated in *Ccl25*^{-/-} mice. These data suggest downregulation of CCR9 accompanied the CD4⁺CD8 α ⁺ T cell differentiation process even in CCL25 independent manner. We have included this important new data in Supplementary Fig. 2d, e and added corresponding discussions on page 14, 15.

R-Fig. 1

(a) Histograms show the CCR9 level among TCRβ⁺CD4⁺CD8α⁺CD8β⁻Foxp3⁻ (CD4⁺CD8αα⁺; red line), TCRβ⁺CD4⁺CD8α⁻CD8β⁻Foxp3⁻ (CD4⁺CD8αα⁻; green line), and TCRβ⁺CD4⁺CD8α⁻CD8β⁻Foxp3⁺ (Tregs; blue line) of IELs in *Ccl25*^{+/+} and *Ccl25*^{-/-} mice. Fluorescence minus one (FMO) control is shown as black line. (b) Graph shows the mean fluorescence intensity (MFI) of CCR9 among CD4⁺CD8αα⁺, CD4⁺CD8αα⁻, Tregs, and FMO control of IELs in *Ccl25*^{+/+} and *Ccl25*^{-/-} mice. Data expressed as mean ± SD of individual mice (n=8, three pooled independent experiments). Statistical analysis was performed by one-way analysis of variance (ANOVA), Turkey test post test (b). **P*<0.05, ***P*<0.01, ****P*<0.001, *****P*<0.0001.

The authors do not provide an explanation as to why their mixed CCR9^{-/-}/WT BM chimera results in Fig 4G and H differ so dramatically from multiple published CCR9^{-/-}/WT mixed BM chimera studies (not just doi.org/10.1002/eji.200535203) that demonstrate that CCR9^{-/-} BM is competitively disadvantaged in generating T cells compared with WT BM. This is because CCR9^{-/-} cells are outcompeted by WT cells in the thymus.

Thank you for your important point regarding the existing literature. We also note that previous studies focused their analysis in the thymus (Uehara S. J Immunol 2002) or spleen (Wurbel MA.

Eur J Immunol 2006), whereas we focused our analysis on intestinal IELs. Nevertheless, to more comprehensively address any discrepancy between our analysis and existing knowledge using *Ccr9*^{-/-} BM chimera settings, we included added two additional more mice and analyzed the CD4⁺ T cells in the spleen, MLN, LPLs, and IELs (R-Fig. 2a). As previously reported, we observed that T cells from *Ccr9*^{-/-} mice were significantly fewer than T cells from *Ccr9*^{+/+} mice at peripheral sites, such as the spleen and MLN, but also among IELs (no significant differences were found in the LPLs, although the trend was similar). Nevertheless, the proportion of CD8αα⁺ in CD4⁺ IELs from *Ccr9*^{-/-} mice was significantly higher than that from *Ccr9*^{+/+} mice (R-Fig. 2b). Hence, these analyses confirm previous publications, while highlighting the different outcomes of T cell differentiation within the gut epithelium. We confirmed that the findings in the spleen were not significantly different from the previous findings. We have included this data in Fig. 2h, and Supplementary Fig. 3o.

R-Fig. 2

(a) Cells were obtained from bone marrow of *Cd45.1⁺Ccr9^{+/+}* mice and *Cd45.2⁺Ccr9^{-/-}* mice and mixed 1:1 ratio. Mixed bone marrow cells were transferred to the lethally irradiated (11Gy) C57B6/J host mice and mice were analyzed 4 weeks after transfer. Graphs shows the frequency of CD45.1⁺ and CD45.2⁺ cells in TCRβ⁺CD4⁺CD8β⁻ SI IELs, LPLs, MLN, and splenocytes. (b) Graphs show the frequency of CD8α⁺ and intracellular Foxp3⁺ among CD45.1⁺TCRβ⁺CD4⁺CD8β⁻ or CD45.2⁺TCRβ⁺CD4⁺CD8β⁻ cells in SI IELs. Data expressed as mean ± SD of individual mice (n=6, three pooled independent experiments). Statistical analysis was performed by student's t test (a, b). **P*<0.05, ***P*<0.01, ****P*<0.001, *****P*<0.0001.

In Figure 3 I am glad to see in the revised manuscript that the authors have repeated the scRNA-seq data and not based their conclusions on comparisons between 1 mouse in each group. Nevertheless, I have several issues with the presentation of the new data. From the presented UMAPs it is difficult to assess alterations in subset proportions (1) between different sites, (2) between CCR9 KO and WT mice and (3) between biological replicates. Instead of the UMAPs in Figure 3B the authors should show bar graphs with proportion of each subset within each of the 4 sites and for each of the individual 6 mice to show how reproducible the data is. The definition of the subsets is also very unclear. What genes were used to define each subset? Why for example are there Tfh cells in the IEL and LPL preps where they would not be expected, and not in the spleen or MLN where they would be expected? Again, in Fig 3D it is impossible to decipher the proportions of each subset between CCR9 KO and WT cells and it is not possible to interpret the data in Fig 3E without the same exact cell numbers being analysed from each site and strain of mouse. The authors need to show cell numbers analysed for each sample and strain. Please present and compare such data in bar graphs with proportions of each subset within a given tissue and mouse and then comparing the three CCR9 KO vs three WT mice but also showing a bar graph for each individual mouse to show reproducibility.

We appreciate the reviewer's comment. To answer the biological replicates and different sites, we added the cell numbers in each sample and the strain (R-Fig. 3a), and the proportion of each

cluster in the IELs and LPLs of each mouse, as shown in Figure 3b, d (R-Fig. 3b, c). In addition, we have added the feature genes related to Figure 3a and c (R-Fig. 4). We appreciate that the new figures make it easy to understand the proportion in *Ccr9*^{+/+} and *Ccr9*^{-/-} IELs. As we agree with the reviewer's point that cells in cluster 9 express the *Tcf7* gene, we have changed the name in cluster 9 and renamed it the *Tcf7*⁺ CD4⁺ T cell cluster. We have included these data in Fig. 3a, Supplementary Fig. 5a, b, Supplementary Fig. 6a, b, figure legend of Fig. 3, and corresponding material and method section on page 24.

a

	IELs	LPLs	MLN	Spleen
Cor9^{+/+} _#1	4843	2256	1106	953
Cor9^{+/+} _#2	1856	681		
Cor9^{+/+} _#3	2097	931		
Cor9^{-/-} _#1	5377	2468	976	615
Cor9^{-/-} _#2	1928	582		
Cor9^{-/-} _#3	2048	810		

b

c

R-Fig.3

Droplet-based scRNA-seq was performed using the Chromium 10X platform. CD4⁺ T cells from splenocytes, MLN cells, LPLs, and IELs were sorted from *Ccr9*^{+/+} and *Ccr9*^{-/-} mice for scRNA-seq. (a) The cell numbers of each sample sequenced. (b) UMAP of scRNA-seq data within each four sites from six individual mice (*Ccr9*^{+/+} and *Ccr9*^{-/-} mice). (c) The proportion of each subset in (b).

R-Fig.4

Cells were sorted by gene expression similarities, and 17 clusters were identified based on to DEGs. Top 20 significantly differentially expressed genes in each cluster were listed. The y-axis displays the $-\log_{10}$ (q-value) for each gene.

Based on the results in Figure 4 the authors make several strong conclusions regarding cellular trajectories and development of CD4⁺CD8⁺ IEL and alterations in KO mice. The conclusions however are based purely on bioinformatic analysis using one trajectory analysis algorithm without direct evidence. The authors should acknowledge this and tone down there conclusions accordingly.

We agreed with the reviewer's comments, so we toned down the discussion section (see page 9 :line 8-9, and page 15 :line21-23).

Minor point

How many mice are presented in Supplementary Figure 6C and were similar observations made for each mouse?

We have compiled all three mice in one figure. We have now added the separated mouse data in the supplemental figure (R-Fig. 5). We conclude that the same TCR expansion occurs in IELs of *Ccr9*^{-/-} mice. We have added the figure into Supplementary Fig. 7d.

R-Fig5

(a) Bar graph representing the ratio of each clonotype among CD4⁺ IELs summed from three animals in each strain. (b) Bar graph representing the ratio of each clonotype among CD4⁺ IELs of individual *Ccr9*^{+/+} and *Ccr9*^{-/-} mice (n=3 in each group).

Reviewer #2 (Remarks to the Author):

The authors have largely addressed my concerns from the prior review. However, in the revision (Supp Fig 3n), the authors state that “cultured T cells from CCR9 KO mice exhibited higher Runx3 and Tbx21 expression than those from CCR9 WT mice.” In this figure, there is a significant reduction in Tbx21 expression between naïve and cultured T cells within the same genotype. No differences between the two genotypes are shown, and thus is there no support for this statement. It is possible that the ex vivo culture conditions may not accurately reflect what the in vivo biology reflected in the scRNAseq data (Fig 5) that does support the authors’ overall conclusion.

Thank you for your comment. We noticed the mislabel in the supplemental figure Y-axis (*Zbtb7b* and *Tbx21* were opposite). which has been corrected in the point-by-point response (R-Fig. 6). We have corrected this figure accordingly. Indeed, the expression level of *Tbx21* in *in vitro* cultured *Ccr9*^{-/-} CD4⁺ T cells was higher than that of *Ccr9*^{+/+} CD4⁺ T cells, as the reviewer mentioned (in IEL-differentiating conditions).

R-Fig. 6

Naïve CD4⁺ T cells were obtained from *Ccr9*^{+/+} and *Ccr9*^{-/-} mice and cultured with anti-CD3/CD28, transforming growth factor- β (TGF- β), retinoic acid (RA), 2,3,7,8-Tetrachlorodibenzodioxin (TCDD) and IFN- γ (IEL-differentiating conditions), with or without CCL25. Three independent experiments were performed in triplicate. Graphs show relative expression of *Runx3*, *Zbtb7b*, and *Tbx21* genes in naïve CD4⁺ T cells and cultured cells of *Ccr9*^{+/+} and *Ccr9*^{-/-} mice. Data are expressed as mean \pm SD of individual experiments. Statistical analysis was performed by one-way ANOVA, Tukey’s post test. * P <0.05, ** P <0.01, *** P <0.001, **** P <0.0001.

In Figure 1a, it appears that there are multiple colored lines behind the FMO in the IEL histograms. These are not in the LPL histograms and may be included in error, otherwise, it is unclear what data those additional lines represent.

We apologize for the multiple colored lines in Figure 1a (R-Fig. 7). This is an error in the manuscript, so we have corrected the figure below.

R-Fig. 7

Histograms show the CCR9 level among TCRβ⁺CD4⁺CD8α⁺CD8β⁻Foxp3⁻ (CD4⁺CD8αα⁺; red line), TCRβ⁺CD4⁺CD8α⁻CD8β⁻Foxp3⁻ (CD4⁺CD8αα⁻; green line), and TCRβ⁺CD4⁺CD8α⁻CD8β⁻Foxp3⁺ (Tregs; blue line) of IELs and LPLs in jejunum and ileum. Fluorescence minus one (FMO) control is shown as black line.

Reviewer #3 (Remarks to the Author):

Thank you for addressing my concerns and providing more clarity and data to the manuscript. I have no more concerns.

We appreciate your comments to improve our manuscript.

REVIEWERS' COMMENTS

Reviewer #1 (Remarks to the Author):

The authors have made a strong effort to address my remaining concerns. I have no further issues with the data and congratulate them on this nice work.

Reviewer #2 (Remarks to the Author):

The authors have sufficiently addressed and made corrections regarding all prior critiques. I thank them for their response and have no further concerns.